# Segment Anything with Robust Uncertainty-Accuracy Correlation

**Hongyou Zhou** [1]  **Marc Toussaint** [1]  **Ling Shao** [2]  **Zihan Ye** [2]

*Figure 1.* **Overview of Robust Uncertainty-Accuracy Correlation (RUAC):** Both panels share the same out-of-domain test images spanning four categories (Objects, Scenes, Scientific, Egocentric). **(a)** Vanilla SAM2 trained on a *fixed* source domain produces *unexplainable* segmentation and *confused* confidence maps, exhibiting *Mask-level Confidence Confusion (MCC)*. **(b)** RUAC introduces Uncertainty Estimation (UE) combined with a *dynamic* training domain via Adversarial Uncertainty Estimation (AUE: bio-inspired style and deformation perturbations), simultaneously improving segmentation accuracy and providing calibrated pixel-wise uncertainty. The result is *Uncertainty-Accuracy Alignment*: *interpretable* segmentation and *robust* uncertainty maps that reliably correlate with prediction errors.

## Abstract

Despite strong zero-shot performance, SAM is unreliable under domain shift due to Mask-level Confidence Confusion (MCC), where a single IoU-based mask score fails to reflect pixel-wise reliability near boundaries. Motivated by the contrast between texture-biased shortcuts in neural networks and shape-centric processing in human vision, we model out-of-domain variation as appearance shifts and non-rigid deformations that jointly perturb images. We propose Segment Anything with **Robust Uncertainty-Accuracy Correlation (RUAC)** for robust pixel-wise uncertainty estimation under appearance and deformation shifts. RUAC adds a lightweight uncertainty head, trains it with a collaborative style-deformation attack that jointly perturbs texture and geometry, and applies Uncertainty-Accuracy Alignment to ensure uncertainty consistently highlights erroneous pixels even under adversarial perturbations. Across 23 zero-shot domains, **RUAC** improves segmentation quality and yields more faithful uncertainty with stronger uncertainty-accuracy correlation. Project page: [hongyouzhou.github.io/ruac](hongyouzhou.github.io/ruac).

[1]Learning and Intelligent Systems, Technical University of Berlin, Berlin, Germany [2]UCAS-Terminus AI Lab, University of Chinese Academy of Sciences, Beijing, China. Correspondence to: Zihan Ye <zihhye@outlook.com>.

*Proceedings of the $43^{rd}$ International Conference on Machine Learning*, Seoul, South Korea. PMLR 306, 2026. Copyright 2026 by the author(s).

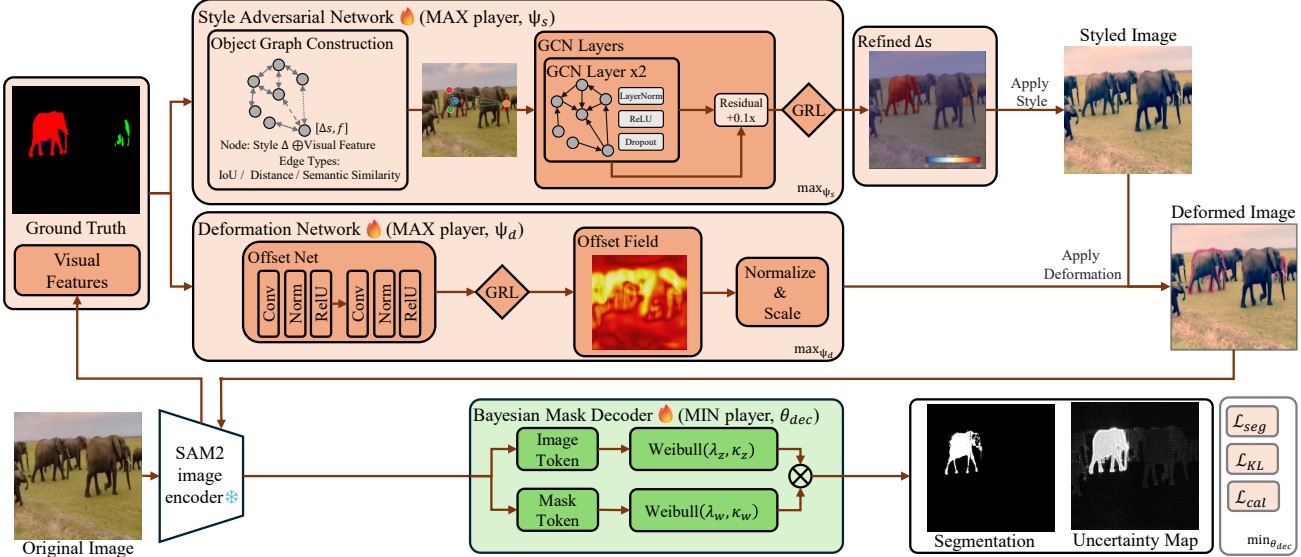

*Figure 2.* **Overview of the RUAC Framework.** We formulate adversarial training as a minimax game between attackers and the segmentation model. The Style Adversarial Network ($\psi_s$) constructs an object graph from ground-truth masks and visual features, then refines per-object style statistics via GCN layers to generate semantically coherent styled images. The Deformation Network ($\psi_d$) predicts a dense offset field from SAM2 visual features to produce geometric perturbations. Both attackers are trained via Gradient Reversal Layers (GRL), enabling end-to-end optimization without PGD-style inner loops. The Bayesian Mask Decoder ($\theta_{dec}$) employs dual-granularity Weibull distributions over image tokens (local, boundary-aware) and mask tokens (global, semantic) to model pixel-wise uncertainty, optimizing for uncertainty-accuracy alignment under these bio-inspired perturbations. Training also includes a clean branch (not shown) to maintain in-domain performance.

## 1. Introduction

Recently, the rise of foundation models has reshaped computer vision. Among them, one of the most striking examples is the Segment Anything Model (SAM) family. From SAM (Kirillov et al., 2023) to the video-optimized SAM2 (Ravi et al., 2025) and concept-aware SAM3 (Carion et al., 2025), these models trained on large-scale datasets (e.g., SA-1B) have demonstrated exceptional zero-shot segmentation across diverse objects and scenes.

Although the SAM family excels at most segmentation tasks, its performance deteriorates on domains outside the pretraining distribution, such as medical imaging (Zhu et al., 2024; Zhang et al., 2024), biological microscopy (Archit et al., 2023), and scientific imaging. This not only leads to a decrease in segmentation accuracy, but we further found that SAM often suffers from **Mask-level Confidence Confusion (MCC)**: Due to the model structure of the SAM family, the entire mask shares the same confidence score, which is hard to differentiate from the background's score. MCC makes it impossible to identify unreliable pixels within a mask, while the uniform confidence across foreground and background renders the model fragile to even minor perturbations. This brittleness is particularly dangerous in safety-critical applications: users cannot reliably determine when the model will fail.

To address MCC, a natural solution is to augment SAM

with a Bayesian mask decoder that provides pixel-level uncertainty estimates rather than mask-level confidence. However, this alone is insufficient: we observe that the learned uncertainty-accuracy relationship, while valid on the source domain, often degrades under domain shift. We term this phenomenon **Uncertainty-Accuracy (UA) shift**: the model's uncertainty no longer correlates with its prediction errors on out-of-domain (OOD) data. Existing fine-tuning approaches (Chen et al., 2024; Wei et al., 2024; Zhu et al., 2024; Zhang et al., 2024) risk catastrophic forgetting, require costly target-domain annotations, and critically, only optimize segmentation metrics (e.g., Dice, IoU) without addressing UA shift (Hendrycks & Dietterich, 2019; Oakden-Rayner et al., 2020; Grote et al., 2024).

To address UA shift, we augment SAM2 with both a Bayesian mask decoder for pixel-level uncertainty and an adversarial training module for uncertainty-accuracy alignment. The result, SAM2 with RUAC (Figure 1), achieves both improved OOD segmentation accuracy and robust uncertainty-accuracy correlation. Unlike prior work that improves either segmentation accuracy or uncertainty calibration, RUAC is, to our knowledge, the first method to simultaneously improve SAM's segmentation accuracy and provide calibrated pixel-wise uncertainty under single-source domain generalization (SDG). The innovation of our method is reflected in three aspects:

1. **Interpretable predictions**: Compared to previous SAM improvements that only focused on accuracy, our method also includes pixel-level uncertainty modeling to identify ambiguous and error-prone regions, allowing users or the model itself to reject or correct predictions with high uncertainty.

2. **Single-source generalization**: Compared to previous fine-tuning-based improvements (Zhang et al., 2024; Archit et al., 2023), our work inherits the philosophy of the SAM family: only training a general segmentation model. This avoids the huge cost of data collection and training in the target data domain, truly strengthening "Segment Anything".

3. **Bio-inspired robustness enhancement**: Compared to previous uncertainty estimation methods, our method is grounded in human vision research: humans exhibit robust object recognition primarily through shape bias, whereas neural networks rely more heavily on texture bias (Geirhos et al., 2019; Landau et al., 1988; Baker et al., 2018; Wang et al., 2025). Therefore, our method enhances robustness from both texture and shape aspects simultaneously, making machine perception consistent with the robustness observed in biological vision and providing robust uncertainty-accuracy correlation. This also brings new insights for future robustness and uncertainty estimation research.

The design of RUAC is motivated by our empirical observations when integrating existing uncertainty methods with SAM2. Specifically, we first follow the fine-tuning settings recommended in SAM2 and evaluate the performance of directly integrating existing uncertainty methods into the SAM2 model. However, the direct integration does not yield satisfactory results: existing uncertainty methods are not designed for foundation models and struggle to generalize when trained on limited fine-tuning data. To address this without requiring additional data, we draw inspiration from human visual perception: humans achieve robust recognition primarily through shape bias rather than texture (Geirhos et al., 2019; Landau et al., 1988; Baker et al., 2018). Accordingly, we decouple OOD variation into two subproblems: style (texture) variation and shape (geometric) variation. While conventional adversarial training targets worst-case prediction error, our approach uses bio-inspired perturbations to improve the correlation between uncertainty and accuracy. Specifically, we propose an adversarial training module that exposes the model to extreme variations in surface appearance and non-rigid deformations, thereby explicitly exploring texture bias and shape understanding capabilities. These variations in surface appearance and non-rigid deformations drive adversarial training to further align uncertainty and error, requiring the model's predictive uncertainty to accurately cover its prediction errors

and uncertainties even under these bio-inspired adversarial samples. This ultimately ensures that when the model fails due to ambiguity in appearance or shape, it can assign high uncertainty to its predictions.

Our contributions are mainly reflected in the following three aspects:

1. We propose a general segmentation framework, RUAC, which exhibits a robust uncertainty-accuracy correlation, truly enhancing the goal of "Segment Anything".

2. We introduce adversarial uncertainty-accuracy alignment: a training paradigm that uses style-deformation perturbations to promote calibration rather than worst-case robustness.

3. We explore existing uncertainty estimation methods under the current foundation model paradigm and demonstrate that they generally perform poorly on out-of-domain data, while our RUAC framework maintains significant robustness in both segmentation accuracy and uncertainty estimation.

## 2. Related Work

**Foundation Models for Segmentation.** SAM (Kirillov et al., 2023) and SAM2 (Ravi et al., 2025) established promptable segmentation as a foundation model paradigm, demonstrating strong transfer across diverse image and video domains. Subsequent works further improve this paradigm for efficiency (Xiong et al., 2024), mask quality (Ke et al., 2023), or specific downstream tasks (Li et al., 2024; Ošep et al., 2024). While domain-specific adaptations (Wei et al., 2024) and robustness-oriented fine-tuning (Chen et al., 2024) have improved performance, the predictive *reliability* of these models under substantial domain shift remains under-explored (Grote et al., 2024).

**Uncertainty and Reliability under Domain Shift.** Deep networks are notoriously miscalibrated (Guo et al., 2017) and overconfident on out-of-domain inputs (Hendrycks & Gimpel, 2017). In dense prediction, this issue is amplified by spatial correlations, necessitating pixel-level uncertainty estimates to pinpoint failure modes. Approaches range from sampling-based Bayesian methods (Gal & Ghahramani, 2016; Kendall & Gal, 2017) to efficient closed-form evidential learning (Sensoy et al., 2018; Ye et al., 2024). Most recently, BNDL (Hu et al., 2025) captures structured uncertainty by modeling latent features as Weibull distributions. Unlike methods that pursue aleatoric-epistemic decomposition, we optimize total predictive uncertainty to align with task error, extending Bayesian methods to target *training-time* reliability under semantically meaningful perturbations.

**Robust Training under Distribution Shift.** Single-source domain generalization (Zhou et al., 2022) employs augmentation (Zhou et al., 2021) or adversarial learning (Ganin & Lempitsky, 2015; Ganin et al., 2016) to mitigate the sensitivity of deep models to texture and style shifts (Geirhos et al., 2019). While $\ell_p$-bounded adversarial training (Goodfellow et al., 2015), particularly Projected Gradient Descent (PGD) (Madry et al., 2018), improves worst-case robustness, it lacks semantic variation. Recent work thus prioritizes meaningful perturbations like style transfer (Geirhos et al., 2019) and geometric deformations (Wang et al., 2021b). Notably, AdvStyle (Zhong et al., 2022) learns adversarial style features for domain generalization, inspiring our learnable style perturbations, while DG-Font (Xie et al., 2021) employs deformable convolutions for feature-level deformations, informing our geometric adversarial module. We leverage these semantic perturbations not primarily for robustness, but to actively align uncertainty with error, ensuring reliable uncertainty estimates under domain shifts.

## 3. Approach

We introduce **RUAC** (**R**obust **U**ncertainty-**A**ccuracy **C**orrelation, Figure 2), a framework that calibrates uncertainty to maintain its semantic relationship with prediction error across domains. It comprises two key components: (1) **Uncertainty Estimation (UE)**: a Bayesian mask decoder that replaces the deterministic mask decoder to produce calibrated uncertainty estimates, and (2) **Adversarial Uncertainty Estimation (AUE)**: an uncertainty-driven adversarial training module that generates challenging samples where the uncertainty-accuracy relationship degrades under domain shift.

### 3.1. Problem Setup

We study *prompted image segmentation* under domain shift. Given an image $\mathbf{I} \in \mathbb{R}^{H \times W \times 3}$ with sparse prompts $\mathbf{P} = \{(x_i, y_i, l_i)\}_{i=1}^n$, where $(x_i, y_i)$ are pixel coordinates and $l_i \in \{+1, -1\}$ indicates positive (foreground) or negative (background) points, a model $\mathcal{M}_\theta$ predicts a binary mask $\hat{\mathbf{M}} \in \{0, 1\}^{H \times W}$ and a pixel-wise uncertainty map $\mathbf{u} \in \mathbb{R}_+^{H \times W}$. Although we build upon SAM2's architecture, we operate in *single-frame mode without memory propagation*: each image is processed independently, isolating segmentation quality from temporal tracking. Training uses a single source domain $\mathcal{D}_s$ (MOSE dataset (Ding et al., 2023), sampling individual frames) and evaluation is zero-shot on $N = 23$ diverse target domains $\{\mathcal{D}_t^i\}_{i=1}^N$ spanning objects, scenes, scientific, and egocentric distributions.

### 3.2. Uncertainty Estimation (UE)

We follow the SAM2 architecture (Ravi et al., 2025) and make one major modification: replacing the deterministic mask decoder with a **Bayesian mask decoder** with parameters $\theta_{\text{dec}}$. Following Hu et al. (2025), we model both image tokens $\mathbf{f} \in \mathbb{R}^{H \times W \times C}$ and mask tokens $\mathbf{m}_k \in \mathbb{R}^C$ using Weibull variational posteriors. We justify the choice of Weibull (vs. Gaussian, Dirichlet, and other non-negative alternatives) in Appendix A.2. For image tokens, a convolutional network predicts spatially-varying per-pixel Weibull parameters $(\lambda_f, \kappa_f)$. For mask tokens, a shared MLP predicts per-channel Weibull parameters $(\lambda_{m,c}, \kappa_{m,c})$ (architecture details in Appendix A). Samples are drawn via reparameterization: $w_i = \lambda_i \cdot (-\ln(1 - u))^{1/\kappa_i}$ where $u \sim \text{Uniform}(0, 1)$. Letting $\mathbf{w}$ denote the stochastic features from both image-token and mask-token branches, we collect all Weibull parameters as $\phi = \{(\lambda_i, \kappa_i)\}$ to form the variational posterior $q_\phi(\mathbf{w})$. Reparameterized samples from both branches are combined via inner product to produce the final logits, enabling closed-form uncertainty propagation where weight uncertainty directly translates to semantic uncertainty in mask predictions. Both image-token and mask-token posteriors contribute to the KL regularization term (Eq. 2), providing calibrated uncertainty at both spatial (per-pixel) and semantic (per-mask) granularities.

**Uncertainty computation.** We support two modes for uncertainty estimation. **(1) Analytic mode:** The Weibull distribution admits closed-form expectation $\mathbb{E}[w_i] = \lambda_i \cdot \Gamma(1 + 1/\kappa_i)$, enabling deterministic forward pass without sampling. Given the analytically-derived logit variance $v$ (Appendix A.8), we apply MacKay's probit approximation (MacKay, 1992) to obtain the expected sigmoid output, then compute the normalized Bernoulli entropy as the pixel-wise uncertainty $\tilde{u}(x, y) \in [0, 1]$. **(2) Monte Carlo mode:** We draw $S$ samples from the Weibull posteriors by sampling $u^{(s)} \sim \text{Uniform}(0, 1)$ and computing reparameterized features via $w_i^{(s)} = \lambda_i \cdot (-\ln(1 - u^{(s)}))^{1/\kappa_i}$ for both image tokens and mask tokens. The mean prediction $\bar{\mathbf{M}}$ and pixel-wise uncertainty (from sample disagreement) are obtained by aggregating across all $S$ samples. While the Bayesian mask decoder provides uncertainty estimates, the learned uncertainty-accuracy relationship may not generalize across domains. This phenomenon, which we term **uncertainty-accuracy (UA) shift**, motivates our adversarial calibration framework.

### 3.3. Adversarial Uncertainty Estimation (AUE)

The AUE module generates challenging samples where the uncertainty-accuracy relationship degrades, exposing failure modes that standard training cannot anticipate. We propose *end-to-end adversarial generation*, where style and deforma-

tion generators are optimized jointly with the segmentation model via Gradient Reversal Layers (GRL) (Ganin et al., 2016). For each training sample we synthesize adversarial images $\mathbf{I}^{\text{adv}}$ via the learnable style and deformation networks described below. The adversarial loss $\mathcal{L}_{\text{adv}} = \mathcal{L}_{\text{seg}}^{\text{adv}} + \beta \mathcal{L}_{\text{KL}}^{\text{adv}}$ reuses the same formulation as the clean branch, ensuring consistent training dynamics while enforcing robustness under domain shifts. In practice, the attack networks are optimized adversarially (via GRL-based sign reversal) to generate such challenging samples, and can additionally be optimized with a dedicated calibration objective (Sec. 3.4) without updating the segmentation model.

**Style adversarial network.** To challenge texture invariance, we introduce a style adversarial network with parameters $\psi_s$, following (Zhong et al., 2022). We first extract per-object style statistics $(\boldsymbol{\mu}_k, \boldsymbol{\sigma}_k) \in \mathbb{R}^3$ as the mean and standard deviation of each RGB channel within the masked region $\mathbf{M}_k$. The network then predicts style residuals $(\Delta\boldsymbol{\mu}_k, \Delta\boldsymbol{\sigma}_k)$ from backbone features $\mathbf{z}$ (coordinated via a GCN for multi-object scenes, see Appendix B.4). Let $\tilde{\boldsymbol{\mu}}_k = \boldsymbol{\mu}_k + \Delta\boldsymbol{\mu}_k$ and $\tilde{\boldsymbol{\sigma}}_k = \boldsymbol{\sigma}_k + \Delta\boldsymbol{\sigma}_k$ denote the perturbed style parameters, and $\tilde{\mathbf{I}}_k = \text{AdaIN}(\mathbf{I}, \tilde{\boldsymbol{\mu}}_k, \tilde{\boldsymbol{\sigma}}_k)$ the stylized image for object $k$. The composite adversarial image is $\mathbf{I}^{\text{adv}} = \mathbf{I}_{\text{bg}} + \sum_k \mathbf{M}_k \odot \tilde{\mathbf{I}}_k$, where $\mathbf{I}_{\text{bg}} = (1 - \sum_k \mathbf{M}_k) \odot \mathbf{I}$.

**Deformation adversarial network.** To challenge shape priors, we introduce a deformation adversarial network with parameters $\psi_d$, inspired by DG-Font (Xie et al., 2021). By design, deformations are small but targeted at boundaries and thin structures, where they yield large segmentation effects analogous to adversarial examples in classification (Szegedy et al., 2014; Goodfellow et al., 2015). The network takes as input the backbone features $\mathbf{z} \in \mathbb{R}^{C \times H \times W}$ and an object mask $\mathbf{M}_k$, then predicts a per-pixel flow field $\boldsymbol{\delta}_k$. Specifically, we first encode the mask via a convolutional downsampler to obtain mask embeddings $\mathbf{e}_k$. These are fused with projected image features via element-wise addition, and passed through a fusion module and offset predictor: $\boldsymbol{\delta}_k = f_{\psi_d}\big(f_{\text{proj}}(\mathbf{z}) + f_{\text{mask}}(\mathbf{M}_k)\big)$. The adversarial image is then obtained via differentiable grid sampling (Jaderberg et al., 2015): $\mathbf{I}^{\text{adv}} = \text{GridSample}(\tilde{\mathbf{I}}, \boldsymbol{\delta})$, where $\tilde{\mathbf{I}}$ is the styled image. Importantly, the same flow field is applied to the ground-truth mask, $\mathbf{M}^{\text{adv}} = \text{GridSample}(\mathbf{M}^*, \boldsymbol{\delta})$, to maintain semantic correspondence between the perturbed image and its supervision signal.

**Perturbation pipeline.** Both style and deformation attacks operate in *image space*: style applies AdaIN color transfer, deformation applies spatial warping via grid sampling. Attack parameters are predicted in *parallel* from the backbone features $\mathbf{z} = f_{\text{enc}}(\mathbf{I})$ of the *original* (unperturbed) image, ensuring stable gradient flow for min-max optimization. These parameters are then *applied sequentially*: style

followed by deformation. A single backbone forward pass on the final adversarial image $\mathbf{I}^{\text{adv}}$ produces adversarial features for decoder training. Perturbation magnitudes are constrained by $\epsilon$ (with $\|\boldsymbol{\delta}\|_\infty \leq \epsilon$ for deformations and analogous bounds for style shifts), ensuring semantically meaningful augmentations (design rationale in Appendix B). Importantly, the attack networks are used *only during training* to expose calibration weaknesses. At inference, only the Bayesian mask decoder runs, requiring no GT masks or attack computation.

### 3.4. Objective and Optimization

**Main objective.** We train the Bayesian mask decoder with segmentation supervision and Bayesian regularization on both clean and adversarial samples. The segmentation loss is defined as

$$\mathcal{L}_{\text{seg}} = \mathcal{L}_{\text{focal}} + \mathcal{L}_{\text{dice}} + \mathcal{L}_{\text{IoU}}, \tag{1}$$

$$\mathcal{L}_{\text{KL}} = \text{KL}\big(q_\phi(\mathbf{w}_{\text{dec}}) \,\|\, p(\mathbf{w}_{\text{dec}})\big), \tag{2}$$

where $\mathcal{L}_{\text{KL}}$ regularizes the Weibull posterior $q_\phi$ towards a Gamma prior $p(\mathbf{w}_{\text{dec}})$ and prevents uncertainty collapse.

Importantly, we do *not* directly supervise the segmentation model parameters using any calibration loss that regresses uncertainty to prediction error. Instead, the main optimizer updates the model only through segmentation supervision and Bayesian regularization on both branches:

$$\min_{\theta_{\text{dec}}} \mathcal{L}_\theta = \underbrace{\mathcal{L}_{\text{seg}} + \beta \mathcal{L}_{\text{KL}}}_{\text{clean branch}} + \gamma \underbrace{\big(\mathcal{L}_{\text{seg}}^{\text{adv}} + \beta \mathcal{L}_{\text{KL}}^{\text{adv}}\big)}_{\text{adversarial branch}}, \tag{3}$$

where $\beta$ controls the KL regularization strength and $\gamma$ weights the adversarial training contribution. We use a curriculum schedule that gradually increases $\gamma$ during training (Appendix C).

**Attacker objective.** Via GRL, the attacker implicitly maximizes the adversarial branch of the total loss. Let $\psi = \{\psi_s, \psi_d\}$ denote attacker parameters, and let $x^{\text{adv}} = T_\psi(x)$ be the adversarial sample produced by composing style and deformation transformations. The attacker objective is:

$$\max_\psi \mathcal{L}_{\text{seg}}^{\text{adv}} + \beta \mathcal{L}_{\text{KL}}^{\text{adv}} + \lambda \mathcal{L}_{\text{cal}}, \tag{4}$$

where $\mathcal{L}_{\text{seg}}^{\text{adv}}$ and $\mathcal{L}_{\text{KL}}^{\text{adv}}$ are the same losses as Eqs. 1–2 evaluated on adversarial samples. The calibration loss $\mathcal{L}_{\text{cal}}$ encourages the model to output uncertainty aligned with prediction errors on adversarial samples.

The calibration loss penalizes both overconfident errors (certain but wrong) and over-uncertain correct predictions (uncertain but correct), each along two complementary chan-

nels:

$$\mathcal{L}_{\text{cal}} = \underbrace{e \cdot \exp(-\text{sg}[u]) + \text{sg}[e] \cdot \exp(-u)}_{\text{certain \& wrong}}$$
$$+ \underbrace{(1 - e) \cdot \exp(\text{sg}[u]) + (1 - \text{sg}[e]) \cdot \exp(u)}_{\text{uncertain \& correct}},$$

(5)

where $e = |\hat{\mathbf{M}} - \mathbf{M}^*|$ is the pixel-wise prediction error, with $\mathbf{M}^*$ denoting the ground-truth mask, and $u$ is the analytic uncertainty. Within each failure mode, the first term provides gradient on the $e$ channel and the second on the $u$ channel. Each stop-gradient $\text{sg}[\cdot]$ routes its term to a single channel. $\text{sg}[u]$ confines the first term to the $e$ channel and blocks the attacker's trivial pathway through $u$. $\text{sg}[e]$ confines the second term to the $u$ channel so it contributes distinct gradient rather than duplicating the $e$-channel signal. The $u$-channel terms are necessary because $\partial \mathcal{L}_{\text{cal}} / \partial e = \exp(-u) - \exp(u)$ vanishes near $u \approx 0$, exactly the regime where the model is wrong but confident. The mirror terms restore gradient signal in this region.

### 3.5. Training Loop

We adopt an *alternating training* scheme where each iteration comprises two forward passes: (1) a *clean pass* on original training samples, and (2) an *adversarial pass* on perturbed samples generated by the style and deformation networks. Both passes share the same segmentation and Bayesian regularization objectives. During the adversarial pass, Gradient Reversal Layers (GRL) implement a min-max update in a single backward pass: segmentation gradients update the encoder/decoder as usual, while their signs are flipped for the attack networks to encourage harder perturbations. This design achieves a min-max optimization without explicit inner loops:

$$\min_{\theta_{\text{dec}}} \max_{\psi_s, \psi_d} \underbrace{\mathcal{L}_{\text{seg}} + \beta\,\mathcal{L}_{\text{KL}}}_{\text{clean branch}} + \gamma \underbrace{\left(\mathcal{L}_{\text{seg}}^{\text{adv}} + \beta\,\mathcal{L}_{\text{KL}}^{\text{adv}} + \lambda\,\mathcal{L}_{\text{cal}}\right)}_{\text{adversarial branch}},$$

(6)

where GRL handles the sign flip for $\psi_s, \psi_d$ implicitly. Algorithm 1 (Appendix) summarizes the complete procedure.

**Interpretation.** From a PAC-Bayesian perspective (McAllester, 1999), our adversarial training can be interpreted as encouraging two desirable properties: (1) *Loss Landscape Flattening*, biasing the posterior toward solutions less sensitive to style and deformation, and (2) *Uncertainty-Risk Coupling*, where regions difficult to make robust tend to exhibit elevated uncertainty (Appendix F.2).

## 4. Experiments

**Datasets:** We evaluate RUAC following the standard zero-shot protocol of SAM/SAM2 (Kirillov et al., 2023; Ravi et al., 2025): a strict SDG setting. Models are fine-tuned solely on the MOSE dataset (Ding et al., 2023) (complex video objects) and evaluated zero-shot on 23 diverse target domains spanning four categories: *Objects*, *Scenes*, *Scientific*, and *Egocentric* (Table S6). This strict SDG protocol tests the model's ability to learn invariant representations rather than memorizing source-specific statistics.

**Prompting protocol (3-click):** For zero-shot evaluation, we follow a deterministic *3-click* interactive prompting protocol consistent with prior promptable/interactive segmentation evaluations (Kirillov et al., 2023; Sofiiuk et al., 2022). Concretely, each object is evaluated with three point prompts generated deterministically from ground truth (with a minimum click separation), and all methods receive identical prompts.

**Metrics:** We report (1) **J&F Score** (Perazzi et al., 2016) for segmentation quality and (2) **PAvPU** (Mukhoti & Gal, 2018) (Patch Accuracy vs. Patch Uncertainty) for uncertainty calibration. All metrics are reported in percentage points.

**Implementation:** Training uses SAM2.1 with the frozen Hiera-B+ image encoder (Hiera Base-Plus, $\sim 69$M parameters). We use AdamW ($10^{-6}$ decoder, $10^{-4}$ uncertainty head/attacker), bfloat16, and a curriculum (Appendix C) over 20 epochs. Training cost, parameter overhead, and a comparison with diffusion-based augmentation are reported in Appendix P. During training, we use analytic uncertainty (Sec. 3.2) with gradients for Bayesian optimization. At evaluation, we report metrics using sampling-based uncertainty ($S$=20 samples) for robustness. Sensitivity to $S$ is analyzed in Appendix O. Full hyperparameters are provided in Appendix Table S5.

**Baselines:** We compare against: (1) *Foundation baselines*: SAM2 zero-shot, SAM2-FT (decoder fine-tuning), and SAM2-FT-LoRA (parameter-efficient fine-tuning (Hu et al., 2022)); (2) *Uncertainty estimation baselines*: Bayes-SAM2 (our Bayesian mask decoder without adversarial training) and UR-ERN (Ye et al., 2024) (evidential uncertainty via NIG head on decoder features; see Appendix E); (3) *Test-time adaptation baseline*: UCTTA (Tan et al., 2025) (uncertainty-calibrated continual test-time adaptation applied to vanilla SAM2 without source-domain fine-tuning), reported separately as it requires model updates during inference; (4) *Generic augmentation baselines*: Random Noise, PGD (Madry et al., 2018), Patch (Nesti et al., 2022), MixStyle (Zhou et al., 2021), DSU (Li et al., 2022), and StyleGen-Generic / StyleGen-Targeted (SD 1.5 img2img following ALIA (Dunlap et al., 2023) and DA-Fusion (Trabucco et al., 2024): Generic uses 2 prompts spanning all target domains, Targeted uses 9 prompts each tailored to one of the 11 below-average OOD domains (overlapping domains merged)), all built on our Bayesian mask decoder.

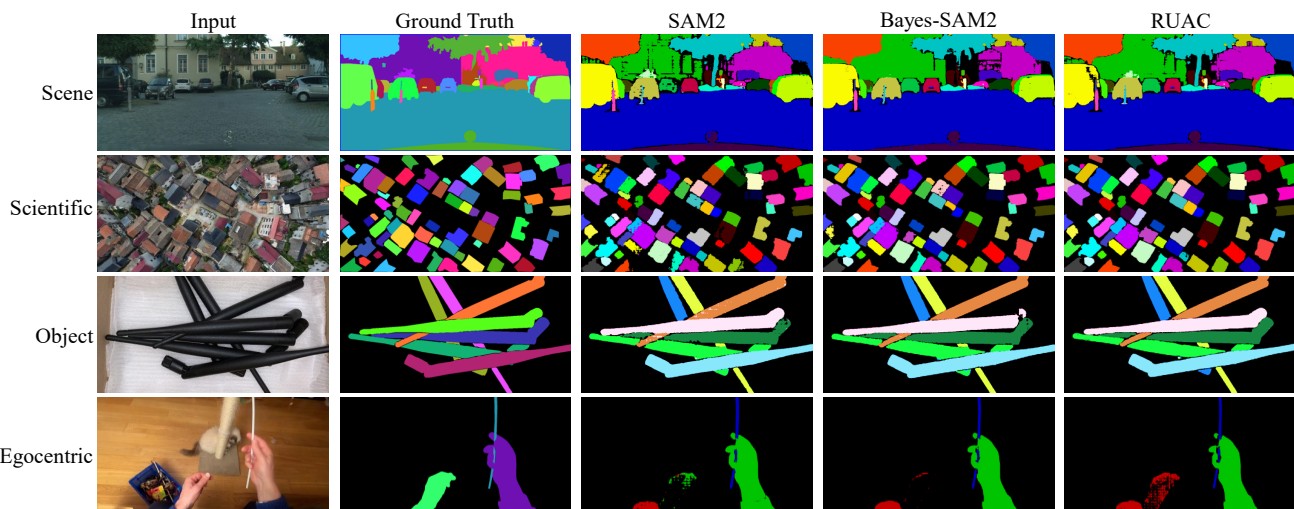

*Figure 3.* **Qualitative segmentation comparison across diverse domains.** From top to bottom: Scene (Cityscapes), Scientific (IBD aerial building imagery), Object (mixed industrial objects), and Egocentric (hand-object interaction). Columns show input, ground truth, and predictions from SAM2, Bayes-SAM2, and RUAC (ours). RUAC produces more complete and accurate masks, particularly for challenging cases such as fine-grained building boundaries (Scene), densely packed structures (Scientific), and occluded objects (Object/Egocentric). Corresponding confidence and uncertainty maps for the first row are shown in Figure 4.

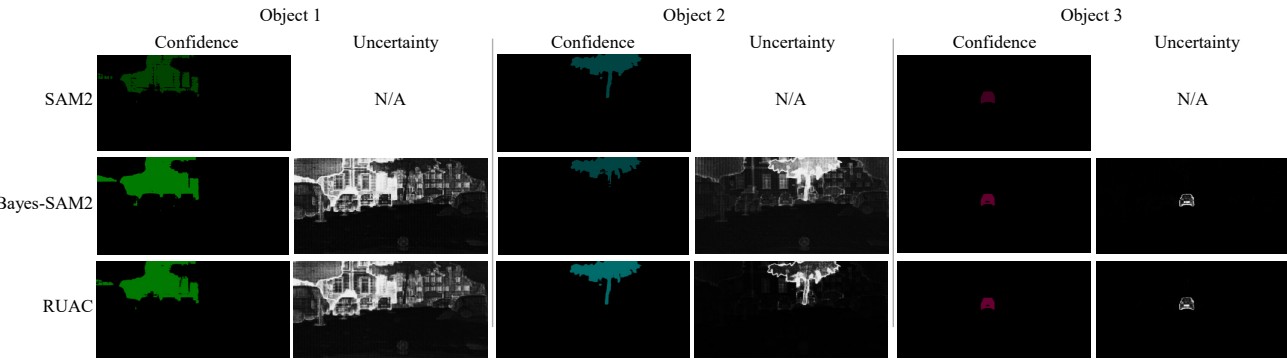

*Figure 4.* **Confidence and uncertainty visualization for three object instances from the first row of Figure 3.** Each row corresponds to a method (SAM2, Bayes-SAM2, RUAC), showing confidence and uncertainty maps. In both maps, brighter colors (greater contrast against the black background) indicate higher confidence or uncertainty. SAM2 produces only confidence without uncertainty estimation (N/A). Both Bayes-SAM2 and RUAC yield uncertainty maps, but RUAC exhibits sharper confidence on object interiors while concentrating uncertainty along ambiguous boundaries, indicating better calibration. Additional visualizations are provided in Figure S7 (Appendix).

## 4.1. Domain Generalization Performance

Table 1 presents the zero-shot performance across all 23 domains. Figure 3 provides qualitative comparisons across four representative domains: RUAC produces more complete masks on urban scenes (finer building boundaries), achieves cleaner delineation of densely packed structures in satellite imagery, and better handles occlusions in object-centric and egocentric views.

**Foundation Baselines:** SAM2 zero-shot yields a strong baseline (Avg 67.75) but struggles on challenging domains like TrashCan (44.9), Hypersim (46.7), and NDISPark (40.7).

**The Impact of RUAC:** RUAC achieves the highest aver-age performance (81.62), surpassing both standard fine-tuning and the Bayesian mask decoder alone. Notably, RUAC shows significant gains on challenging datasets (e.g., PIDRay +6.1 vs. SAM2), where texture and shape priors differ most from natural images.

## 4.2. Uncertainty Calibration Analysis

A key motivation of RUAC is aligning uncertainty with risk. While standard fine-tuning (SAM2-FT) often produces "con-fident errors" (high confidence, wrong prediction) on OOD data, RUAC maintains well-calibrated uncertainty estimates. This is qualitatively evident in Figure 4, which visualizes confidence and uncertainty maps for three object instances from the first row of Figure 3. In both maps, brighter col-

*Table 1.* Zero-shot domain generalization results (J&F scores) on 23 target datasets. All methods are fine-tuned on MOSE and evaluated zero-shot. Bayes-SAM2 denotes UE only. RUAC denotes full framework with UE and AUE. **Bold** and underline indicate the best and second-best results per column (excluding ablations and test-time adaptation).

| | Objects | | | | | | | | Scenes | | | | | | | Scientific | | | | Egocentric | | | | |
| Method | TrashCan | iShape | ZeroWaste-f | LVIS | NDD20 | Plittersdorf | TimberSeg | DRAM | Cityscapes | ADE20K | Hypersim | WoodScape | STREETS | NDISPark | DOORS | BBBC038v1 | IBD | PIDRay | PPDLS | GTEA | EgoHOS | VISOR | OVIS | Avg↑ |
|---|---|---|---|---|---|---|---|---|---|---|---|---|---|---|---|---|---|---|---|---|---|---|---|---|
| *Foundation baselines (no uncertainty estimation)* | | | | | | | | | | | | | | | | | | | | | | | | |
| SAM2 | 44.9 | 63.3 | 83.2 | 75.2 | 91.7 | 47.5 | 78.3 | 70.0 | 64.2 | 58.8 | 46.7 | 55.6 | 87.6 | 40.7 | 89.1 | 81.1 | 80.9 | 63.4 | 76.2 | 55.6 | 84.0 | 52.0 | 68.3 | 67.75 |
| SAM2-FT | 72.4 | 79.1 | 87.3 | 75.9 | 92.3 | 86.2 | 83.8 | **78.6** | 65.1 | **60.2** | 60.8 | 59.1 | 88.5 | 86.7 | 91.6 | 87.2 | 87.6 | 68.3 | 83.7 | 89.6 | 89.3 | 74.7 | 86.3 | 79.75 |
| SAM2-FT-LoRA | 71.3 | 78.0 | 86.8 | 75.6 | 93.8 | 88.7 | 83.6 | 78.5 | 61.6 | 51.1 | 54.6 | 59.6 | 89.8 | 88.2 | **92.2** | 87.0 | 88.9 | 64.4 | 83.6 | 90.5 | 90.4 | 73.0 | **88.5** | 79.13 |
| *Uncertainty estimation baselines (no adversarial calibration)* | | | | | | | | | | | | | | | | | | | | | | | | |
| Bayes-SAM2 | 74.9 | 84.3 | 88.0 | 75.1 | 94.2 | 88.4 | 88.3 | 73.6 | 55.4 | 55.2 | 57.5 | 57.0 | 90.4 | 89.6 | 90.9 | 85.5 | 90.3 | 67.5 | 86.7 | 91.0 | 90.4 | 75.1 | 87.7 | 79.87 |
| UR-ERN | 68.5 | 62.8 | 83.4 | 66.0 | 89.6 | 83.3 | 69.2 | 76.6 | 58.2 | 53.8 | 54.4 | 53.7 | 86.0 | 83.3 | 90.4 | 81.6 | 80.6 | 62.2 | 76.6 | 85.1 | 72.1 | 67.2 | 83.6 | 73.40 |
| *Generic augmentation baselines (with Bayesian mask decoder)* | | | | | | | | | | | | | | | | | | | | | | | | |
| Random Noise | 74.4 | 84.6 | 88.4 | 74.8 | 94.5 | 88.6 | 88.3 | 76.0 | 64.2 | 56.4 | 61.8 | 58.3 | 90.4 | 89.7 | 91.9 | 87.5 | 90.2 | 66.5 | 86.9 | 91.3 | 91.3 | 74.8 | 87.8 | 80.81 |
| PGD | 72.7 | 84.6 | 87.5 | 73.2 | 94.3 | 88.4 | 88.2 | 75.8 | 53.3 | 56.0 | 55.8 | 59.4 | 90.5 | 89.3 | 91.6 | 88.4 | 90.1 | 67.4 | 87.5 | 91.1 | 91.5 | 74.0 | 87.3 | 79.90 |
| Patch | 73.2 | 84.7 | 88.0 | 75.4 | 94.3 | 88.9 | 88.3 | 76.1 | 61.3 | 54.4 | 60.9 | 59.1 | 90.3 | 89.4 | 91.8 | 86.0 | 90.3 | 67.7 | 87.4 | 90.8 | 91.2 | 76.7 | 87.7 | 80.60 |
| MixStyle | 75.9 | 85.4 | 87.9 | 77.2 | 94.6 | 88.9 | 88.4 | 70.5 | 59.9 | 53.4 | 59.9 | 60.6 | 90.6 | 89.2 | 91.0 | 87.7 | 89.8 | 66.5 | 86.7 | 90.8 | 91.9 | 73.9 | 88.1 | 80.37 |
| DSU | 73.7 | 84.6 | 88.4 | 79.5 | 94.6 | 89.0 | 88.6 | 73.9 | 58.6 | 51.8 | 57.8 | 59.6 | 90.6 | 88.9 | 91.2 | 88.2 | 90.3 | 67.1 | 87.2 | 90.5 | 90.5 | 75.8 | 87.9 | 80.35 |
| StyleGen-Generic | 76.5 | 80.6 | 87.5 | 71.8 | 94.5 | 88.3 | 86.7 | 74.5 | 62.0 | 47.9 | 51.2 | 61.3 | 90.8 | 88.9 | 91.4 | 88.2 | 88.3 | 68.6 | 86.4 | 89.9 | 90.3 | 71.1 | 87.9 | 79.33 |
| StyleGen-Targeted | 75.8 | 78.1 | 83.7 | 74.7 | 93.3 | 87.3 | 85.0 | 75.8 | 51.9 | 51.3 | 48.1 | 59.9 | 90.4 | 88.8 | 91.7 | 89.2 | 84.9 | 71.6 | 86.6 | 89.5 | 87.2 | 67.7 | 85.9 | 78.19 |
| *Adversarial augmentation ablations (Style & Deform)* | | | | | | | | | | | | | | | | | | | | | | | | |
| RUAC-Style-Global | 71.3 | 84.9 | 88.6 | 72.5 | 94.4 | 88.6 | 88.7 | 72.6 | 60.0 | 55.1 | 62.6 | 58.8 | 90.4 | 88.6 | 90.9 | 88.3 | 90.4 | 68.1 | 86.4 | 91.6 | 90.5 | 75.3 | 87.3 | 80.26 |
| RUAC-Style-Single | 75.0 | 84.6 | 88.3 | 74.6 | 94.3 | 88.8 | 88.4 | 75.0 | 58.3 | 56.9 | 59.8 | 59.5 | 90.4 | 89.4 | 90.9 | 85.8 | 90.0 | 67.2 | 87.7 | 90.2 | 91.6 | 74.9 | 88.2 | 80.43 |
| RUAC-Style-Multi | 71.0 | 85.4 | 88.0 | 75.0 | 94.7 | 88.7 | 88.4 | 75.3 | 61.7 | 55.3 | 58.8 | 59.2 | 90.3 | 89.7 | 91.9 | 88.1 | 90.5 | 67.2 | 87.6 | 90.9 | 91.1 | 75.0 | 88.1 | 80.52 |
| RUAC-Style-MultiBG | 74.3 | 86.3 | 88.6 | 74.2 | 94.4 | 88.6 | 88.7 | 75.8 | 59.3 | 55.0 | 62.5 | 60.9 | 90.3 | 88.6 | 91.8 | 87.8 | 90.2 | 67.6 | 87.3 | 90.0 | 91.0 | 75.0 | 87.3 | 80.68 |
| RUAC-Style-GCN | 74.4 | 85.7 | 88.2 | 74.7 | 94.0 | 88.4 | 88.8 | 75.6 | 60.5 | 57.3 | 62.2 | 59.4 | 90.3 | 88.9 | 91.1 | 87.8 | 90.1 | 68.5 | 87.3 | 90.1 | 91.2 | 75.4 | 87.2 | 80.74 |
| RUAC-Deform | 71.0 | 85.7 | 88.7 | 74.4 | 94.5 | 88.8 | 88.4 | 74.9 | 60.6 | 54.7 | 60.7 | 58.7 | 90.3 | 88.7 | 91.4 | 88.3 | 90.6 | 68.2 | 85.8 | 91.4 | 91.2 | 75.9 | 87.1 | 80.44 |
| *Complete methods (ours)* | | | | | | | | | | | | | | | | | | | | | | | | |
| RUAC | 73.8 | **86.1** | **89.1** | 78.6 | **94.6** | 88.6 | **88.7** | 74.9 | **66.7** | 58.5 | **64.0** | **61.5** | 90.4 | 89.1 | 91.5 | 87.7 | **90.6** | 69.5 | 87.0 | **91.4** | 91.4 | 76.6 | 86.8 | **81.62** |
| *Test-time adaptation (for reference only, since it unfairly uses test data)* | | | | | | | | | | | | | | | | | | | | | | | | |
| UCTTA | 73.6 | 84.9 | 88.5 | 80.6 | 94.7 | 87.6 | 90.1 | 78.6 | 70.1 | 60.8 | 66.3 | 60.9 | 90.0 | 86.2 | 91.9 | 86.9 | 89.8 | 67.5 | 88.5 | 87.5 | 91.7 | 77.6 | 87.5 | 81.82 |

*Table 2.* **PAvPU Comparison.** Patch Accuracy vs. Patch Uncertainty across domains. Higher PAvPU indicate better uncertainty-accuracy alignment.

| | In-Domain | Out-of-Domain | | | | | | | | | | | | | | | | | | | | | | | |
| Method | MOSE | TrashCan | iShape | ZeroWaste | LVIS | NDD20 | Plittersdorf | TimberSeg | DRAM | Cityscapes | ADE20K | Hypersim | WoodScape | STREETS | NDISPark | DOORS | BBBC | IBD | PIDRay | PPDLS | GTEA | EgoHOS | VISOR | OVIS | Avg↑ |
|---|---|---|---|---|---|---|---|---|---|---|---|---|---|---|---|---|---|---|---|---|---|---|---|---|---|
| Bayes-SAM2 | **86.9** | 61.5 | 67.2 | 69.4 | 64.9 | 29.5 | 94.6 | 15.2 | 11.3 | 25.6 | 31.3 | 72.1 | 83.7 | 95.6 | 99.7 | 64.6 | 58.7 | 53.9 | 9.0 | 43.6 | 64.7 | 44.1 | 33.3 | 81.2 | 55.4 |
| RUAC | 81.6 | 70.5 | 78.3 | 83.0 | 71.5 | 60.3 | 99.8 | 26.4 | 10.8 | 30.9 | 37.9 | 74.2 | 82.6 | 97.0 | 99.8 | 70.6 | 71.4 | 58.5 | 14.9 | 51.4 | 73.7 | 63.5 | 51.9 | 86.9 | **63.7** |

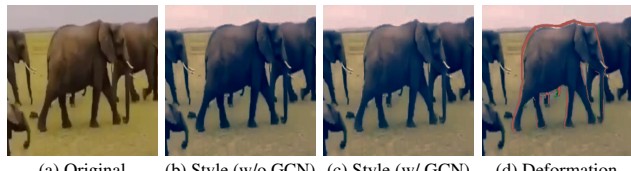

| (a) Original | (b) Style (w/o GCN) | (c) Style (w/ GCN) | (d) Deformation |

*Figure 5.* **Visualization of adversarial perturbations (zoomed to central region).** (a) Original input image. (b) Without GCN, style perturbation is applied globally, causing uniform darkening across the entire scene. (c) With GCN capturing inter-object correlations, perturbations become object-aware, enabling more targeted attacks while preserving global semantics (e.g., the central elephant). (d) Geometric deformation, where red indicates reduced regions and green indicates expanded regions along object boundaries.

ors (greater contrast against the black background) indicate higher values. SAM2 provides only confidence estimates without uncertainty quantification. Both Bayes-SAM2 and RUAC produce uncertainty maps, but RUAC exhibits sharper confidence on object interiors while concentrating uncertainty along ambiguous boundaries, better tracking prediction reliability under domain shift.

**PAvPU Analysis:** To further evaluate uncertainty calibration, we use the Patch Accuracy vs. Patch Uncertainty (PAvPU) metric (Mukhoti & Gal, 2018). PAvPU measures the proportion of samples that are either correctly predicted with low uncertainty (accurate-certain) or incorrectly predicted with high uncertainty (inaccurate-uncertain). Following the BNDL evaluation protocol (Hu et al., 2025), we average PAvPU across three uncertainty thresholds $\tau \in \{0.01, 0.05, 0.1\}$. Higher values indicate better uncertainty-accuracy alignment. Table 2 reports these results for all evaluation domains. Critically, a random uncertainty estimator (independent of prediction accuracy) achieves $\sim 50$ PAvPU. Bayes-SAM2's OOD PAvPU of 55.40 is *barely above chance*, indicating that its uncertainty calibration largely collapses under domain shift, a manifestation of the

*Table 3.* **Calibration comparison across augmentation methods using the same Bayesian mask decoder.** Aggregate metrics over 23 OOD datasets. **Bold** and underline mark the best and second-best per metric (excluding the no-augmentation Bayes-SAM2). RUAC is the only method that improves both segmentation accuracy and all three calibration metrics over Bayes-SAM2.

| Method | J&F↑ | PAvPU↑ | AURC↓ | ECE↓ |
|---|---|---|---|---|
| Bayes-SAM2 | 79.87 | 55.40 | 0.102 | 0.123 |
| Random Noise | 80.81 | 52.24 | 0.112 | 0.126 |
| PGD | 79.90 | 52.89 | 0.115 | 0.130 |
| Patch | 80.60 | 56.29 | 0.105 | 0.131 |
| MixStyle | 80.37 | 55.46 | 0.100 | 0.133 |
| DSU | 80.35 | 58.02 | 0.097 | 0.125 |
| StyleGen-Generic | 79.33 | 55.10 | 0.101 | 0.136 |
| StyleGen-Targeted | 78.19 | 48.87 | 0.111 | 0.172 |
| RUAC | **81.62** | **63.72** | **0.092** | **0.117** |

UA shift problem. In contrast, RUAC achieves 63.72 OOD PAvPU, a **+8.32 absolute** improvement over Bayes-SAM2. On in-domain data, both methods perform well (Bayes-SAM2 86.9 vs. RUAC 81.6), but the OOD gap reveals RUAC's robustness to domain shift. Notably, on some OOD datasets where the model correctly identifies domain shift, it expresses elevated uncertainty even when predictions are accurate. We term this *domain-level cautiousness*, a conservative behavior desirable in safety-critical applications. While PAvPU is threshold-sensitive and may underestimate performance when uncertainty is globally elevated, threshold-free metrics confirm that uncertainty *ranking* remains meaningful: RUAC achieves 0.81 $AUROC_{pixel}$ (failure detection at pixel granularity) and 0.59 Uncertainty-Accuracy PCC on OOD datasets, indicating that even within high-uncertainty regimes, the model correctly ranks which pixels are more likely to fail (details in Appendix H). Additionally, AURC analysis (Appendix I) shows RUAC achieves 9.8% lower risk at matched coverage levels compared to Bayes-SAM2, and Wilcoxon signed-rank tests (Appendix J) confirm statistical significance ($p < 0.001$ for AURC, $p < 0.01$ for PAvPU) across 23 domains. Expected Calibration Error (ECE) further confirms calibration quality: RUAC achieves 0.117 versus 0.123 for Bayes-SAM2 across 23 domains (17/23 wins, $p = 0.037$, Appendix J).

**Comparison with generic augmentation baselines.** Across seven generic augmentation methods built on the same Bayesian mask decoder (Tables 1, 3), RUAC is the only one that simultaneously improves J&F and all three calibration metrics (PAvPU, AURC, ECE) over the no-augmentation Bayes-SAM2. The mechanism is that RUAC alone perturbs the most segmentation-relevant feature directions at sufficient magnitude (Appendix K).

### 4.3. Ablation Study

Table 1 includes ablations of our adversarial generators: **Style-Global:** global style perturbation applied uniformly across the image. **Style-Single:** style perturbation applied only to the target object. **Style-Multi:** style perturbation applied to multiple objects independently. **Style-MultiBG:** multi-object style perturbation including background. **Style-GCN:** multi-object with GCN refinement for semantically coherent perturbations. **Deform:** deformation attack only (no style). **RUAC:** combines Style-GCN with Deform, yielding the best generalization by probing complementary directions (texture vs. shape). Figure 5 visualizes these perturbations: GCN enables object-aware style transfer while deformation probes geometric robustness. RUAC is robust to $\beta$ (KL regularization) and $\gamma$ (adversarial weight). Full sensitivity sweeps are reported in Appendix L, and a loss-component ablation is provided in Appendix M.

### 4.4. Downstream Utility: Uncertainty-Guided Mask Correction

To test whether calibration translates into downstream utility, we apply a simple uncertainty-guided connected-component filter (UncCorr) that removes mask fragments with high mean uncertainty, applied identically to Bayes-SAM2 and RUAC (Table 4). UncCorr *improves* RUAC's J&F (+0.09, 17/23 datasets) but *degrades* Bayes-SAM2's ($-0.28$, 14/23 datasets): miscalibrated uncertainty actively misleads downstream filters, while RUAC's calibrated uncertainty enables productive post-processing. Implementation details are in Appendix N.

*Table 4.* **Uncertainty-guided mask correction.** J&F before/after UncCorr filtering, with Wins/Losses over 23 OOD datasets.

| Method | J&F↑ | + UncCorr (Δ) | Wins / Losses |
|---|---|---|---|
| Bayes-SAM2 | 79.87 | 79.59 ($-0.28$) | 9 / 14 |
| **RUAC** | **81.62** | **81.71** (+0.09) | **17 / 6** |

## 5. Conclusion

We presented RUAC, a framework that addresses UA shift via bio-inspired adversarial training. RUAC achieves reliable zero-shot transfer across 23 OOD datasets. By aligning pixel uncertainty with prediction error, RUAC outperforms passive uncertainty methods, demonstrating that adversarial calibration is key to trustworthy segmentation under domain shift. Beyond calibration, RUAC's uncertainty maps are actionable for downstream mask correction (Appendix N). Future work includes extending RUAC to Vision-Language Models for trustworthy multi-modal reasoning, to future SAM versions such as SAM3, and to video tracking for uncertainty-aware temporal consistency.

## Impact Statement

This paper presents work whose goal is to advance the field of Machine Learning. There are many potential societal consequences of our work, none which we feel must be specifically highlighted here.

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

# Supplementary Material

## Table of contents:

# A. Bayesian Mask Decoder Architecture and Design Rationale

We adapt the Bayesian Non-negative Decision Layer (BNDL) (Hu et al., 2025) for SAM2's mask decoder. Our implementation differs from the original in several key aspects, each motivated by the unique requirements of promptable segmentation with uncertainty estimation.

## A.1. Uncertainty Quantification, AU/EU, and the Relation to BNDL

Uncertainty quantification (UQ) decomposes uncertainty into aleatoric uncertainty (AU, the inherent randomness in the data) and epistemic uncertainty (EU, the uncertainty inherent in the model itself) (Gal & Ghahramani, 2016). BNDL (Hu et al., 2025) maps these two sources to two Weibull-parameterized streams: the latent representation $z$ models AU and the decision-layer weights $w$ model EU. Predictions are obtained by inner product of reparameterized samples from both streams.

Our adaptation for SAM2 changes how each Weibull stream is conditioned while preserving the AU/EU correspondence. (i) The AU stream replaces BNDL's per-sample $\kappa$ with a small CNN that predicts per-pixel $\kappa$, making the $z$-Weibull spatially heterogeneous as required for dense prediction (Sec. A.4). (ii) The EU stream replaces BNDL's static decision-layer weight with a dynamic weight projected from each of SAM2's $K$ mask tokens, giving every mask hypothesis its own Weibull posterior over decision weights (Sec. A.6). The $\kappa$-prediction network for $w$ keeps BNDL's original structure. Both streams remain Weibull-parameterized and combine via inner product as in BNDL, so the AU/EU decomposition is structurally preserved.

Our uncertainty estimation is based on BNDL because it is a lightweight, state-of-the-art UQ method with strong interpretability, fitting our goal of *interpretably segment anything* under zero-shot domain shifts. Although our adaptation preserves BNDL's AU/EU structure, we do not pursue further improvement on AU and EU estimations. The more important question for our task is *how to guarantee robust uncertainty-accuracy correlation under domain shift*, an orthogonal goal to AU/EU decomposition.

This motivates our framework: Segment Anything with Robust Uncertainty-Accuracy Correlation (RUAC). RUAC retains the Weibull-based Bayesian decoder of BNDL for its interpretable per-pixel uncertainty and augments it with a bio-inspired adversarial training objective (Sec. 3.3) that explicitly aligns predictive uncertainty with segmentation error under domain shift. To our knowledge, RUAC is the first method that leverages uncertainty-based adversarial training to simultaneously improve SAM's segmentation accuracy and provide calibrated pixel-wise uncertainty under single-source domain generalization (SDG). The remainder of this section details our adaptation of BNDL for SAM2's mask decoder.

## A.2. Weibull Posterior Modeling

**Design choice:** We parameterize both pixel features and mask weights with Weibull distributions $W(\lambda, \kappa)$ supported on $[0, \infty)$.

**Rationale:** Although BNDL was originally proposed for classification, its structural requirement transfers naturally to our segmentation setting: BNDL models non-negative evidence for $C$ classes per image, while our setting requires non-negative evidence for $K = 4$ mask hypotheses per pixel (with ReLU-activated features through STE-ReLU). Both reduce to modeling non-negative evidence for discrete categories, for which a distribution supported on $[0, \infty)$ is the natural choice. Weibull additionally admits closed-form expectation $\mathbb{E}[w] = \lambda \cdot \Gamma(1 + 1/\kappa)$ and variance (Sec. A.8), enabling analytic uncertainty propagation through the inner-product logit computation without sampling overhead. Common alternatives are inadequate for the following reasons:

- **Gaussian**: has support on $(-\infty, +\infty)$ and assigns non-zero probability to negative evidence values, violating the non-negativity constraint.
- **Dirichlet**: commonly used in Evidential Deep Learning, models distributions over probability simplices rather than continuous evidence vectors, requiring an extra layer of indirection that Weibull avoids.
- **Other non-negative distributions**: among distributions with support on $[0, \infty)$, Weibull subsumes Exponential as a special case ($\kappa = 1$) and offers simpler closed-form moments than Log-Normal or Gamma.

- **Truncated Gaussian**: can enforce non-negativity but breaks the closed-form moment structure required for analytic uncertainty propagation, forcing reliance on Monte Carlo estimation.

## A.3. Dual-Path Architecture

**Design choice:** Unlike the original BNDL which processes only pixel features, our version uses two parallel paths: (1) a **pixel feature path** processing upscaled decoder features ($C' = 32$ channels), and (2) a **mask token path** processing SAM's $K$ learned mask tokens ($K = 4$: 1 single + 3 multi-mask). **Rationale:** SAM's multi-mask output requires each mask hypothesis to have *independent* uncertainty characteristics. A single global uncertainty estimate cannot capture the confidence differences between competing mask hypotheses (e.g., when the prompt is ambiguous). By processing mask tokens through a separate Weibull-parameterized path, each hypothesis maintains its own $(\lambda, \kappa)$ parameters, enabling the model to express "mask A is confident but mask B is uncertain" for the same pixel location. Note that this dual-path design is motivated by SAM's multi-mask output structure (per-hypothesis uncertainty), not by AU/EU separation. We treat uncertainty as a unified signal for reliability assessment, as discussed in Sec. A.1.

## A.4. Spatial Kappa Prediction

**Design choice:** While the original BNDL uses a per-sample scalar $\kappa$ for the AU stream (one $\kappa$ per image, predicted by a Linear-Softplus head from the pooled feature), we predict spatially-varying $\kappa$ using a small CNN (Conv$_{3 \times 3}$-GELU-Conv$_{3 \times 3}$-Softplus), clamped to $[0.5, 10.0]$. **Rationale:** Segmentation uncertainty is inherently *spatially heterogeneous*. Object boundaries, occluded regions, and ambiguous textures should exhibit higher uncertainty than confident interior regions. A per-image scalar $\kappa$ forces the model to choose a single confidence level for the entire image, which is inappropriate for dense prediction tasks. The convolutional architecture enables $\kappa$ to vary smoothly across space while respecting local image context. The clamping bounds prevent numerical instability (very small $\kappa$ causes heavy tails, while very large $\kappa$ collapses to point estimates).

## A.5. STE-ReLU Activation

**Design choice:** We use Straight-Through Estimator (STE) ReLU (Bengio et al., 2013) (hard ReLU in forward, identity gradient in backward) instead of smooth alternatives like Softplus. **Rationale:** In Evidential Deep Learning (EDL), zero evidence must map to maximum uncertainty. Softplus never produces exact zeros, meaning the model cannot express "I have no evidence about this pixel". STE-ReLU produces true sparsity in the forward pass (exact zeros where evidence is absent) while avoiding the dead neuron prob-

lem through gradient flow in the backward pass. This is critical for calibrated uncertainty: regions with zero foreground evidence should have entropy 0.5 (maximum binary uncertainty), not a slightly-below-maximum value.

## A.6. Mask Token Processing

**Design choice:** Each mask token is processed through a shared MLP that outputs per-channel Weibull parameters for foreground/background classes ($2 \times C'$ dimensions). The final logits compute $\text{logits}_k = \langle \tilde{z}, \tilde{w}_{k,\text{fg}} \rangle - \langle \tilde{z}, \tilde{w}_{k,\text{bg}} \rangle + b_k$. **Rationale:** The foreground-background formulation mirrors SAM's binary mask prediction while enabling probabilistic interpretation. The inner-product formulation ensures that uncertainty propagates correctly: if either pixel features or weights are uncertain (high variance Weibull samples), the resulting logits will have high variance, correctly reflecting the compound uncertainty.

## A.7. Training vs. Inference Mode

**Design choice:** During training, we sample from the Weibull distribution via reparameterization. At inference, we use the analytic expectation $\mathbb{E}[\tilde{z}] = \lambda \cdot \Gamma(1 + \kappa^{-1})$. **Rationale:** Sampling during training is necessary for gradient-based optimization of the Weibull parameters. However, at inference, using the analytic expectation provides deterministic predictions (same input $\Rightarrow$ same output), which is preferable for reproducibility and downstream applications. The analytic expectation is also more efficient (no sampling overhead) and has lower variance than any finite sample estimate.

## A.8. Weibull Variance Computation

The Bayesian mask decoder parameterizes both pixel features $z_c$ and per-mask weights $w_{k,c}$ with Weibull distributions. Each Weibull variable with parameters $(\lambda, \kappa)$ has closed-form variance:

$$\text{Var}[X] = \lambda^2 \left[ \Gamma\left(1 + \frac{2}{\kappa}\right) - \Gamma^2\left(1 + \frac{1}{\kappa}\right) \right]. \quad (S7)$$

The final logit is computed as an inner product between pixel features and mask weights: $\ell_k = \sum_c z_c \cdot w_{k,c}$. For two *independent* random variables, the variance of their product is:

$$\text{Var}[ZW] = \text{Var}[Z]\text{Var}[W] + \text{Var}[Z]\mathbb{E}[W]^2 + \text{Var}[W]\mathbb{E}[Z]^2. \quad (S8)$$

We use a first-order approximation that retains only the $\text{Var}[Z]\text{Var}[W]$ term:

$$v \approx \sum_c \text{Var}[z_c] \cdot \text{Var}[w_{k,c}]. \quad (S9)$$

**Justification for this approximation:** The omitted terms $\text{Var}[Z]\mathbb{E}[W]^2$ and $\text{Var}[W]\mathbb{E}[Z]^2$ scale the variance by a roughly constant factor across pixels. Our empirical analysis confirms:

1. **Rank preservation**: The Pearson correlation between simplified and full variance exceeds **0.99** across diverse inputs. Since calibration metrics (AUROC, PAvPU) depend only on the *ranking* of uncertainty values, not their absolute magnitudes, the approximation preserves all task-relevant information.

2. **Constant scaling**: The full variance is approximately $2$–$3\times$ the simplified variance. This constant factor is absorbed by the subsequent MacKay approximation, which maps variance to probability through $\kappa = 1/\sqrt{1 + \pi v/8}$. A constant scaling of $v$ merely shifts the sigmoid operating point uniformly, without affecting relative ordering.

3. **Analytic-sampling agreement**: Comparing our analytic uncertainty against 100-sample Monte Carlo ground truth yields Pearson correlation $>$ **0.90**, confirming that the end-to-end pipeline (variance approximation + MacKay) produces well-ranked uncertainty estimates.

This variance $v$ is then used in MacKay's probit approximation to compute analytic uncertainty.

# B. Adversarial Network Design Decisions

**Pipeline overview.** Both style and deformation attacks operate in *image space*: style applies AdaIN color transfer, deformation applies spatial warping via grid sampling. Attack parameters are predicted in *parallel* from the backbone features $\mathbf{z} = f_{\text{enc}}(\mathbf{I})$ of the *original* (unperturbed) image, ensuring stable gradient flow for min-max optimization. These parameters are then *applied sequentially* in the order style→deformation. A single backbone forward pass on the final adversarial image produces adversarial features for decoder training. This design minimizes backbone forward passes (2 per iteration: clean + adversarial) while maintaining cooperative attack dynamics.

## B.1. Optimization Strategy

We adopt Gradient Reversal Layers (GRL) for minimax optimization rather than PGD-style inner loops or alternating optimization steps. Traditional alternating optimization (e.g., updating attacker for $k$ steps, then defender for $k$ steps) introduces instability from non-stationary objectives. GRL achieves equivalent minimax dynamics within a single backward pass: the segmentation model receives gradients that minimize loss, while attack networks receive negated gradients that maximize loss, enabling simultaneous parameter updates with consistent gradient information. Note that we still perform both clean and adversarial forward passes each iteration (see Algorithm 1), but the min-max update itself requires no inner optimization loop.

## B.2. Style Adversarial Network

For appearance perturbations, we use Adaptive Instance Normalization (AdaIN) rather than direct pixel perturbations or neural style transfer. Direct perturbations (e.g., PGD) create high-frequency noise that models can easily filter out, while not reflecting realistic domain shifts. AdaIN operates on per-channel statistics (mean, variance), corresponding to realistic appearance changes like lighting, color grading, and material properties. This forces genuine generalization to different visual appearances rather than simple noise filtering.

Per-object features are extracted via masked average pooling: $\mathbf{f}_k = \frac{\sum_{x,y} \mathbf{z}_t(x,y) \cdot \mathbf{M}_k(x,y)}{\sum_{x,y} \mathbf{M}_k(x,y) + \epsilon}$. An MLP predicts style residuals $\Delta \boldsymbol{s}_k = \text{GRL}(\text{MLP}(\mathbf{f}_k))$, constrained via sigmoid-based relative encoding:

$$\boldsymbol{\mu}_k^{\text{adv}} = \boldsymbol{\mu}_k \cdot (1 + \epsilon_\mu \cdot (2\sigma(\Delta \boldsymbol{s}_k^\mu) - 1)) + \epsilon \cdot \tanh(\Delta \boldsymbol{s}_k^{\text{shift}}), \tag{S10}$$

$$\boldsymbol{\sigma}_k^{\text{adv}} = \boldsymbol{\sigma}_k \cdot (1 + \epsilon_\sigma \cdot (2\sigma(\Delta \boldsymbol{s}_k^\sigma) - 1)), \tag{S11}$$

where $\epsilon_\mu, \epsilon_\sigma$ control perturbation magnitude.

When multiple objects are present ($K \geq 2$), an optional Graph Convolutional Network coordinates perturbations across objects. Independent per-object perturbations may create physically inconsistent scenes (e.g., conflicting lighting conditions). The GCN refines style residuals based on spatial proximity and visual similarity, with edges $e_{ij} = \mathbb{1}[d_{\text{spatial}}(i, j) < \tau_d] \cdot \mathbb{1}[\text{sim}_{\text{visual}}(\mathbf{f}_i, \mathbf{f}_j) > \tau_s]$. A 2-layer GCN with symmetric normalization produces coordinated, scene-consistent perturbations.

## B.3. Deformation Adversarial Network

The deformation network reuses SAM2's memory encoder architecture (MaskDownSampler, CXBlock Fuser) rather than a custom design. This architecture is already optimized for fusing mask information with image features, which is precisely what semantic deformation prediction requires. Reusing it reduces additional parameters, leverages pretrained fusion capabilities, and ensures matching receptive fields.

Given backbone features and object masks, the network predicts dense offset fields:

$$\mathbf{h}_k = \text{Fuser}(\text{Proj}(\mathbf{z}_t) + \text{MaskEnc}(\mathbf{M}_k)), \tag{S12}$$

$$\Delta \mathbf{p}_k = \text{GRL}(\epsilon \cdot (2\sigma(\text{OffsetNet}(\mathbf{h}_k)) - 1)), \tag{S13}$$

where $\Delta \mathbf{p}_k \in \mathbb{R}^{2 \times H \times W}$ and $\epsilon$ bounds maximum displacement. The adversarial image is then obtained via differen-

tiable grid sampling: $\mathbf{I}^{\text{adv}} = \text{GridSample}(\tilde{\mathbf{I}}, \boldsymbol{\delta})$, where $\tilde{\mathbf{I}}$ is the styled image from the previous stage.

Several design choices ensure training stability. (1) Deformation offsets are zero-initialized so that the attack network produces identity transformations initially, allowing the model to first learn basic segmentation before encountering deformations. (2) Encoder components (MaskEnc, Proj, Fuser) are frozen to prevent GRL gradients from destabilizing pre-trained modules. (3) Zero-mean offsets are enforced by subtracting the spatial mean, preserving local warps while removing global translation.

**Ground-truth mask synchronization.** Crucially, the same flow field $\boldsymbol{\delta}$ is applied to the ground-truth mask via bilinear interpolation: $\mathbf{M}^{\text{adv}} = \text{GridSample}(\mathbf{M}^*, \boldsymbol{\delta})$. This ensures semantic correspondence between the deformed image and its supervision signal, enabling correct loss computation on the adversarial branch. For multi-object scenes, each object mask is warped independently using its corresponding offset field and composited via weighted averaging in overlap regions.

### B.4. Graph Convolutional Network for Multi-Object Coordination

When multiple objects are present, independent per-object perturbations may create physically inconsistent scenes. We employ a 2-layer Graph Convolutional Network (GCN) to coordinate style perturbations across spatially or semantically related objects.

**Graph Construction.** Given $K$ object masks $\{\mathbf{M}_k\}_{k=1}^{K}$ and their visual features $\{\mathbf{f}_k\}_{k=1}^{K}$ (extracted via masked average pooling from backbone features), we construct an object graph $\mathcal{G} = (\mathcal{V}, \mathcal{E})$ where each node represents an object. Edges are established based on three complementary criteria:

1. **Spatial overlap (IoU):** Objects with overlapping masks should have coordinated perturbations:

$$w_{ij}^{\text{IoU}} = \text{IoU}(\mathbf{M}_i, \mathbf{M}_j) \cdot \mathbb{1}[\text{IoU}(\mathbf{M}_i, \mathbf{M}_j) > \tau_{\text{IoU}}]. \quad \text{(S14)}$$

2. **Geometric proximity:** Nearby objects likely share lighting conditions:

$$w_{ij}^{\text{dist}} = \max\left(0, 1 - \frac{d_{\text{boundary}}(\mathbf{M}_i, \mathbf{M}_j)}{d_{\max}}\right) \cdot \mathbb{1}[d_{ij} < \tau_d], \quad \text{(S15)}$$

where $d_{\text{boundary}}$ is the minimum boundary distance between masks.

3. **Semantic similarity:** Visually similar objects should receive similar perturbations:

$$w_{ij}^{\text{sem}} = \cos(\mathbf{f}_i, \mathbf{f}_j) \cdot \mathbb{1}[\cos(\mathbf{f}_i, \mathbf{f}_j) > \tau_{\text{sim}}]. \quad \text{(S16)}$$

The final edge weight combines all criteria: $w_{ij} = w_{ij}^{\text{IoU}} + w_{ij}^{\text{dist}} + w_{ij}^{\text{sem}}$. Self-loops with unit weight are added to preserve node identity.

**Node Feature Initialization.** Each node is initialized with the concatenation of style residuals and projected visual features:

$$\mathbf{h}_k^{(0)} = [\Delta\boldsymbol{\mu}_k, \Delta\boldsymbol{\sigma}_k] \oplus \text{MLP}_{\text{proj}}(\mathbf{f}_k) \in \mathbb{R}^{12}, \quad \text{(S17)}$$

where the MLP projects 256-dimensional visual features to 6 dimensions, balancing influence with the 6-dimensional style residuals.

**Message Passing.** We use row-normalized graph convolution with residual connections:

$$\mathbf{H}^{(l+1)} = \text{ReLU}\left(\text{LayerNorm}\left(\tilde{\mathbf{D}}^{-1}\tilde{\mathbf{A}}\mathbf{H}^{(l)}\mathbf{W}^{(l)}\right)\right), \quad \text{(S18)}$$

where $\tilde{\mathbf{A}} = \mathbf{A} + \mathbf{I}$ is the adjacency matrix with self-loops, $\tilde{\mathbf{D}}_{ii} = \sum_j \tilde{\mathbf{A}}_{ij}$ is the degree matrix, and $\mathbf{W}^{(l)}$ are learnable weights. The final layer projects back to style dimension without activation.

**Residual Output.** To ensure stable training with GRL, refined style residuals use a scaled residual connection:

$$[\Delta\boldsymbol{\mu}', \Delta\boldsymbol{\sigma}'] = [\Delta\boldsymbol{\mu}, \Delta\boldsymbol{\sigma}] + \alpha \cdot \mathbf{H}_{[:6]}^{(L)}, \quad \text{(S19)}$$

where $\alpha = 0.1$ is a fixed scaling factor that prevents large initial perturbations while allowing gradual refinement.

## C. Training Strategy and Curriculum

### C.1. Hardware Configuration

All experiments are conducted on $8\times$ NVIDIA A40 GPUs (48GB each) using PyTorch's Distributed Data Parallel (DDP). We use a per-GPU batch size of 2 with gradient accumulation of 2 steps, yielding an effective batch size of 32. Mixed-precision training (bfloat16) is enabled for memory efficiency. Total training time for 20 epochs on MOSE is approximately 6 hours.

### C.2. Curriculum Schedule

Training follows a three-phase curriculum to ensure stable learning. In Phase 1 (epochs 0-4), only the standard SAM segmentation loss is active ($\gamma = 0$), establishing a working baseline. Phase 2 (epochs 4-6) introduces Bayesian uncertainty estimation ($\beta = 0.05$). Phase 3 (epochs 6+) enables full AUE with adversarial training ($\gamma = 0.2$). This progressive hardening prevents destabilization from adversarial perturbations before the model has acquired basic segmentation capability.

## C.3. Parameter Configuration

The image encoder (Hiera-B+) remains frozen throughout training. SAM's encoder was trained on billions of masks with carefully designed pretraining. Fine-tuning on our smaller MOSE dataset risks catastrophic forgetting of general visual features and overfitting to source-specific appearances. Freezing preserves general-purpose representations while the decoder adapts to uncertainty-aware segmentation.

We use a learning rate hierarchy: decoder ($10^{-6}$), uncertainty head and attacker networks ($10^{-4}$). Randomly initialized components (uncertainty head, attackers) require faster learning to catch up with the pre-trained decoder, while the decoder updates slowly to preserve pre-trained knowledge.

*Table S5.* Hyperparameter summary.

| Category | Hyperparameter | Value |
|---|---|---|
| Training | Epochs | 20 |
| | Effective batch size | 32 |
| | Optimizer | AdamW |
| Learning rates | Image encoder | frozen |
| | Mask decoder | $10^{-6}$ |
| | Bayesian mask decoder | $10^{-4}$ |
| | Attackers | $10^{-3} \to 10^{-4}$ |
| Loss weights | $\beta$ (UE) | 0.05 |
| | $\gamma$ (AUE) | 0.2 |
| | $\lambda$ (calibration) | 0.1 |
| Adversarial | Style $\epsilon$ | 0.3 |
| | Deform $\epsilon$ | 0.15 |
| Bayesian mask decoder | Feature dim $C'$ | 32 |
| | $\kappa$ range | $[0.5, 10.0]$ |
| | KL weight | $10^{-11}$ |
| | Prior $\Gamma(\alpha, \beta)$ | $(1.0, 3.0)$ |

## C.4. How Calibration Loss Updates Each Channel

The dual stop-gradient in Eq. 5 routes calibration gradient through two complementary pathways. We trace each.

**Step 1: $e$ channel trains the attacker to find miscalibrated samples.** The terms with $\text{sg}[u]$ flow gradient through the prediction error $e$, back through the sigmoid and decoder to the adversarial image $\mathbf{I}^{\text{adv}}$. Gradient Reversal flips the sign for attacker parameters $\psi_s, \psi_d$, so the attacker maximizes $\mathcal{L}_{\text{cal}}$ by generating perturbations that expose calibration failures: samples where the model is either (a) confident but wrong (CW), or (b) uncertain but correct (UC).

**Step 2: $u$ channel trains the Bayesian decoder to align $u$ with $e$.** The mirror terms with $\text{sg}[e]$ flow gradient through the analytic uncertainty $u$ into the Weibull parameters of the Bayesian mask decoder. The gradient drives $u$ upward at high-$e$ pixels (CW corner) and downward at low-$e$ pixels (UC corner), shaping the posterior toward calibration. The same gradient also reaches the attacker via GRL, encourag-

**Algorithm 1** RUAC Training via Gradient Reversal

---

**Require:** Image batch $\{\mathbf{I}\}$, GT masks $\{\mathbf{M}^*\}$, attack networks $\mathcal{A}_{\text{style}}, \mathcal{A}_{\text{deform}}$
1: Initialize encoder $f_{\text{enc}}$, Bayesian mask decoder $f_{\text{dec}}$, optimizer $\mathcal{O}$
2: **for** each iteration **do**
3:     *// Clean forward pass*
4:     $\mathbf{z} \leftarrow f_{\text{enc}}(\mathbf{I}); \hat{\mathbf{M}}, \tilde{u} \leftarrow f_{\text{dec}}(\mathbf{z})$
5:     $\mathcal{L}_{\text{clean}} \leftarrow \mathcal{L}_{\text{seg}} + \beta\mathcal{L}_{\text{KL}}$
6:     *// Adversarial forward pass (GRL reverses gradients for attackers)*
7:     *// Predict attack parameters in parallel from clean features* $\mathbf{z}$
8:     $(\Delta\boldsymbol{\mu}, \Delta\boldsymbol{\sigma}) \leftarrow \mathcal{A}_{\text{style}}(\mathbf{z}, \mathbf{M}^*); \boldsymbol{\delta} \leftarrow \mathcal{A}_{\text{deform}}(\mathbf{z}, \mathbf{M}^*)$
9:     *// Apply attacks sequentially: style $\to$ deformation*
10:     $\tilde{\mathbf{I}} \leftarrow \text{AdaIN}(\mathbf{I}, \boldsymbol{\mu} + \Delta\boldsymbol{\mu}, \boldsymbol{\sigma} + \Delta\boldsymbol{\sigma})$
11:     $\mathbf{I}^{\text{adv}} \leftarrow \text{GridSample}(\tilde{\mathbf{I}}, \boldsymbol{\delta})$
12:     $\mathbf{z}^{\text{adv}} \leftarrow f_{\text{enc}}(\mathbf{I}^{\text{adv}}); \hat{\mathbf{M}}^{\text{adv}}, \tilde{u}^{\text{adv}} \leftarrow f_{\text{dec}}(\mathbf{z}^{\text{adv}})$
13:     $e \leftarrow |\hat{\mathbf{M}}^{\text{adv}} - \mathbf{M}^*|$
14:     $\mathcal{L}_{\text{adv}} \leftarrow \mathcal{L}_{\text{seg}}^{\text{adv}} + \beta\mathcal{L}_{\text{KL}}^{\text{adv}} + \lambda\mathcal{L}_{\text{cal}}$
15:     *// Joint update: GRL flips gradients for $\psi_s, \psi_d$*
16:     $\mathcal{L}_{\text{total}} \leftarrow \mathcal{L}_{\text{clean}} + \gamma\mathcal{L}_{\text{adv}}$
17:     $\mathcal{O}.\text{step}(\nabla_{\theta,\phi,\psi}\mathcal{L}_{\text{total}})$     *(GRL reverses $\nabla_\psi$)*
18: **end for**

---

ing it to generate samples with mismatched $(e, u)$ patterns the model has not yet learned to handle.

**Step 3: Decoder trains on the resulting hard samples.** The main model additionally sees these adversarial samples through $\mathcal{L}_{\text{seg}}^{\text{adv}} + \beta\mathcal{L}_{\text{KL}}^{\text{adv}}$. The segmentation loss provides error signal, the KL term regularizes the Bayesian posterior, and the $u$-channel signal from Step 2 directly aligns uncertainty with error. Together, training in regions selected by the attacker forces the model into parameter regions where uncertainty aligns with prediction risk, the "flat minima" effect from PAC-Bayesian theory (Section F).

**Why dual stop-gradient prevents trivial solutions.** A naive calibration loss that lets gradients flow freely between $e$ and $u$ would invite two failure modes:

1. **Uniform uncertainty collapse**: the model could minimize the loss by raising $u$ everywhere, ignoring whether $e$ is actually large.
2. **Attacker $u$-shortcut**: the attacker could maximize the loss by manipulating $u$ via BNDL rather than producing genuinely hard inputs.

The dual stop-gradient blocks both shortcuts. $\text{sg}[e]$ in the $u$-channel terms makes $u$-updates conditional on the actual error: $u$ is only pushed up where $e$ is observed to be high, preventing uniform collapse. $\text{sg}[u]$ in the $e$-channel terms keeps the attacker's adversarial signal flowing through real prediction error rather than through $u$. Each stop-gradient closes the partner channel's shortcut while leaving the legitimate gradient path open.

*Table S6.* Zero-shot target datasets (23 total). All models train on MOSE only.

| Category | Datasets |
|---|---|
| Objects | TrashCan (Hong et al., 2020), iShape (Yang et al., 2021), ZeroWaste-f (Bashkirova et al., 2022), LVIS (Gupta et al., 2019), NDD20 (Trotter et al., 2020), Plittersdorf (Haucke et al., 2022), TimberSeg (Fortin et al., 2022), DRAM (Cohen et al., 2022) |
| Scenes | Cityscapes (Cordts et al., 2016), ADE20K (Zhou et al., 2019), Hypersim (Roberts et al., 2021), WoodScape (Yogamani et al., 2019), STREETS (Snyder & Do, 2019), NDISPark (Ciampi et al., 2021; 2022), DOORS (Pugliatti & Topputo, 2022) |
| Scientific | BBBC038v1 (Caicedo et al., 2019), IBD (Chen et al., 2022), PIDRay (Wang et al., 2021a), PPDLS (Minervini et al., 2016) |
| Egocentric | GTEA (Fathi et al., 2011; Li et al., 2015), EgoHOS (Zhang et al., 2022), VISOR (Darkhalil et al., 2022; Damen et al., 2022), OVIS (Qi et al., 2022) |

## C.5. Why GT Masks During Training Do Not Cause Overfitting

A potential concern is that using ground-truth masks to generate adversarial perturbations might cause the model to overfit to the training data. We address this concern:

**Training-only component.** The attack networks $\mathcal{A}_{\text{style}}$ and $\mathcal{A}_{\text{deform}}$ are used *exclusively during training*. At inference, only the Bayesian mask decoder runs, requiring no GT masks or attack networks. This follows the standard adversarial training paradigm (Madry et al., 2018).

**GT defines *where* to perturb, not *what* to predict.** The GT masks serve a different role than in supervised learning: they localize objects for perturbation generation, not as prediction targets. The decoder learns to maintain calibration under perturbations, not to memorize GT patterns.

**Zero-shot generalization validates no overfitting.** Our evaluation is conducted on 23 *completely held-out* OOD datasets (zero-shot). The consistent improvements (Table 1, 2) demonstrate effective generalization, not memorization of source-domain GT patterns.

## D. Dataset Details

**Source domain:** MOSE (Ding et al., 2023) contains 2,149 video clips with 5,200 objects across 36 categories. We use *first frames only* for both training and evaluation, operating SAM2 in single-frame mode without memory propagation. This isolates image segmentation quality from temporal tracking, enabling fair comparison with image-based methods. **Target domains:** Table S6 shows the 23 zero-shot datasets spanning medical imaging (BBBC038v1),

autonomous driving (Cityscapes, WoodScape), industrial inspection (ZeroWaste-f, PIDRay), indoor scenes (Hypersim, DOORS), and egocentric settings (EgoHOS, VISOR). For datasets originally provided as videos, we evaluate on first frames only. **Prompting protocol (3-click).** For zero-shot evaluation, we follow a deterministic *3-click* interactive prompting protocol consistent with prior promptable/interactive segmentation evaluations (Kirillov et al., 2023; Sofiiuk et al., 2022). Each object is evaluated with three point prompts generated deterministically from ground truth (with a minimum click separation). All methods receive identical prompts.

## E. UR-ERN Baseline Implementation

UR-ERN (Ye et al., 2024) is an evidential deep learning method that models predictive uncertainty via Normal Inverse Gamma (NIG) distributions. To adapt UR-ERN for SAM2, we add a lightweight $1 \times 1$ convolutional head on the upscaled decoder features (32 channels after transposed convolutions). This head predicts four channels corresponding to the NIG parameters $(\gamma, \nu, \alpha, \beta)$, which are constrained to valid domains via softplus activations ($\nu > 0$, $\alpha > 1$, $\beta > 0$) and linear activation for $\gamma \in \mathbb{R}$.

Following the original formulation, we decompose the predictive variance into aleatoric and epistemic components: $\sigma^2_{\text{aleatoric}} = \beta/(\alpha - 1)$ and $\sigma^2_{\text{epistemic}} = \beta/(\nu(\alpha - 1))$. The NIG negative log-likelihood and evidence regularization terms are used for training. We report total predictive variance as the uncertainty measure.

## F. Theoretical Analysis: Why AUE Aligns Uncertainty with Error

In this section, we provide a theoretical perspective on why optimizing a Bayesian mask decoder under bounded adversarial perturbations can improve the alignment between predictive uncertainty and model error. We ground this analysis in the Variational Inference (VI) framework.

### F.1. Preliminaries

Let $\mathcal{D}$ be the data distribution. We seek a posterior $q_\phi(\mathbf{w})$ over weights $\mathbf{w}$ that minimizes the Variational Free Energy (equivalent to maximizing the Evidence Lower Bound, ELBO):

$$\mathcal{L}(\phi) = \mathbb{E}_{\mathbf{x},\mathbf{y}\sim\mathcal{D}} \left[ \mathbb{E}_{\mathbf{w}\sim q_\phi}[-\log p(\mathbf{y}|\mathbf{x},\mathbf{w})] \right] + \beta \, \text{KL}(q_\phi(\mathbf{w})\|p(\mathbf{w})). \tag{S20}$$

In AUE, we augment this with an adversarial objective. Let $\delta^*(\mathbf{x})$ be the worst-case perturbation within a feasible set $\Delta$ (defined by our Style and Deformation generators):

$$\delta^*(\mathbf{x}) = \arg\max_{\delta\in\Delta} \mathbb{E}_{\mathbf{w}\sim q_\phi}[-\log p(\mathbf{y}|\mathbf{x}_\delta,\mathbf{w})]. \tag{S21}$$

The AUE objective effectively minimizes the risk on these perturbed samples:

$$\mathcal{L}_{\text{AUE}}(\phi) \approx \mathbb{E}_{\mathbf{x},\mathbf{y}} \left[ \mathbb{E}_{\mathbf{w} \sim q_\phi}[-\log p(\mathbf{y}|\mathbf{x}_{\delta^*}, \mathbf{w})] \right] + \beta \text{KL}(q_\phi \| p). \tag{S22}$$

### F.2. Logit Variance Increases Predictive Entropy

We first formalize a key property of our analytic uncertainty estimator. Note that the logit variance $v$ is strictly non-negative by construction, ensured by the Weibull parameterization. Under the MacKay-style sigmoid approximation, increased logit variance produces higher predictive entropy.

**Lemma F.1.** *Let $m \in \mathbb{R}$ and $v \geq 0$ denote the mean and variance of a (scalar) logit, and define the MacKay/probit approximation*

$$p(v) \triangleq \sigma(\kappa_s(v)\, m), \quad \kappa_s(v) \triangleq (1 + cv)^{-1/2}, \quad c > 0. \tag{S23}$$

*Let $u(v) \triangleq H(p(v))$ be the Bernoulli entropy $H(p) = -p \log p - (1-p)\log(1-p)$. Then $u(v)$ is non-decreasing in $v$. Moreover, if $m \neq 0$ then $u(v)$ is strictly increasing in $v$.*

*Proof.* Define the effective logit magnitude $a(v) \triangleq \kappa_s(v)\,|m| = |m|\,(1 + cv)^{-1/2}$. Note that for $m \neq 0$, $a(v)$ is strictly decreasing in $v$ (and constant if $m = 0$).

Due to the symmetry of entropy $H(p) = H(1-p)$, we have $u(v) = H(\sigma(|m|\kappa_s(v))) = H(\sigma(a(v)))$. Let $g(a) \triangleq H(\sigma(a))$ for $a \geq 0$. Differentiating $g(a)$ with respect to $a$:

$$\frac{\mathrm{d}g}{\mathrm{d}a} = \underbrace{\frac{\mathrm{d}H}{\mathrm{d}p}\bigg|_{p=\sigma(a)}}_{-\text{logit}(p)=-a} \cdot \underbrace{\frac{\mathrm{d}\sigma}{\mathrm{d}a}(a)}_{\sigma(a)(1-\sigma(a))} = -a \cdot \sigma(a)(1 - \sigma(a)). \tag{S24}$$

For $a > 0$, we have $\sigma(a) \in (0.5, 1)$, so both terms are positive, making $\mathrm{d}g/\mathrm{d}a$ strictly negative. Thus, $u(v)$ is the composition of a decreasing function $g(a)$ and a decreasing function $a(v)$, making it strictly increasing in $v$ (for $m \neq 0$). $\square$

### F.3. Connection to Flat Minima and Uncertainty

We now provide an intuition for why optimizing Eq. S22 improves uncertainty calibration. The core argument relies on the connection between adversarial robustness and the geometry of the loss landscape (Flat Minima).

**Proposition F.2** (Informal). *Adversarial training acts as a smoothing operator on the loss landscape, encouraging convergence to regions with lower curvature (flatter minima) compared to standard training. In a variational framework, lower curvature in the loss $\mathcal{L}$ w.r.t weights $\mathbf{w}$ permits the posterior covariance $\Sigma$ to expand (increase) without incurring high likelihood penalties. Through Lemma F.1, this*

*increased posterior variance translates to higher predictive entropy.*

*Proof Sketch.* Let us approximate the posterior $q_\phi(\mathbf{w})$ as a Gaussian $\mathcal{N}(\boldsymbol{\mu}, \boldsymbol{\Sigma})$. Minimizing the free energy (Eq. S20) involves a trade-off between fitting the data (minimizing NLL) and staying close to the prior (minimizing KL). Using a second-order Taylor expansion of the loss around $\boldsymbol{\mu}$, the optimal covariance $\boldsymbol{\Sigma}^*$ satisfies the inverse-Hessian relationship:

$$\boldsymbol{\Sigma}^* \approx (\mathbf{H}_{\mathbf{w}} + \lambda \mathbf{I})^{-1}, \tag{S25}$$

where $\mathbf{H}_{\mathbf{w}}$ is the Hessian of the loss and $\lambda$ corresponds to the prior precision.

**1. Standard Training (Sharp Minima):** Standard ERM often converges to sharp minima where the Hessian $\mathbf{H}_{\mathbf{w}}$ has large eigenvalues (high curvature). To minimize the expected NLL term $\mathbb{E}_w[\text{NLL}] \approx \text{NLL}(\boldsymbol{\mu}) + \frac{1}{2}\text{Tr}(\mathbf{H}(\boldsymbol{\Sigma}))$, the optimizer is forced to reduce $\boldsymbol{\Sigma}$ significantly. This results in small posterior variance, low logit variance, and consequently overconfident predictions (low entropy).

**2. AUE Training (Flat Minima):** Optimizing against perturbations $\delta^*$ effectively minimizes the worst-case loss within a neighborhood, which necessitates finding a solution $\boldsymbol{\mu}$ that is robust to local changes. This implies finding a region where the loss surface is flat (low curvature, small $\mathbf{H}_{\mathbf{w}}$). Because $\mathbf{H}_{\mathbf{w}}$ is smaller, the penalty term $\text{Tr}(\mathbf{H}(\boldsymbol{\Sigma}))$ is reduced, allowing the KL term to dominate. The KL term encourages $\boldsymbol{\Sigma}$ to expand towards the prior (unit variance). This larger $\boldsymbol{\Sigma}$ propagates to higher logit variance, which, via Lemma F.1, results in higher predictive entropy (calibrated uncertainty).

**Conclusion:** AUE aligns uncertainty with error by forcing the model into flat minima. In these regions, the Bayesian posterior is allowed to express higher epistemic uncertainty (weight variance), which correctly reflects the model's reliability, whereas sharp minima force the posterior to collapse into overconfidence.

### F.4. Generalization Bound

From a PAC-Bayesian viewpoint, the AUE objective can be viewed as maximizing the margin of validity for the posterior. By training on $\mathbf{x}_{\delta^*}$, we essentially optimize the risk over a smoothed distribution. Scalable PAC-Bayes bounds often contain terms relating to the sharpness of the posterior. By enabling the posterior to be flatter (higher variance) while maintaining low empirical risk, AUE effectively optimizes a tighter bound on potential OOD error.

## G. Failure Cases and Limitations

We observe three main failure modes: (1) **Extreme domain shift**: X-ray and microscopy images degrade segmentation

quality, though uncertainty remains useful for failure detection. (2) **Very small objects**: Objects <1% of image area challenge both segmentation and uncertainty estimation due to limited spatial resolution of uncertainty maps. (3) **Heavy occlusion**: Objects >80% occluded produce appropriate high uncertainty but may fail to segment correctly. In such cases the model correctly "knows that it doesn't know" but cannot recover the missing information. These limitations motivate future work on multi-scale uncertainty estimation and targeted adversarial attacks for edge cases.

# H. Threshold-Free Uncertainty Analysis

While PAvPU provides a threshold-based calibration metric, it can be sensitive to the choice of uncertainty threshold. On some OOD datasets, the model may express elevated uncertainty across most pixels, correctly identifying domain shift, which leads to low PAvPU even when predictions are accurate. This conservative behavior is desirable in safety-critical applications.

To provide a more complete picture, we report threshold-free metrics that assess whether uncertainty correctly *ranks* predictions by error likelihood, independent of any specific threshold choice.

## H.1. Failure Detection AUROC

Area Under the Receiver Operating Characteristic ($AUROC_{pixel}$) measures the probability that a randomly chosen *incorrectly predicted pixel* has higher uncertainty than a randomly chosen *correctly predicted pixel*. An AUROC of 0.5 indicates random ranking, while 1.0 indicates perfect separation. This is evaluated at **pixel granularity**: each pixel serves as an independent sample.

As shown in Figure S6, RUAC achieves an average $AUROC_{pixel}$ of **0.809** across all OOD datasets. Even on datasets with low PAvPU (e.g., DRAM, PIDRay), $AUROC_{pixel}$ remains above 0.7, indicating that uncertainty correctly identifies which pixels are more likely to be mis-predicted.

## H.2. Uncertainty-Accuracy Pearson Correlation

Pearson Correlation Coefficient (PCC) between pixel-wise uncertainty and pixel-wise error provides a continuous measure of calibration. Positive correlation indicates that higher uncertainty corresponds to higher error rates.

RUAC achieves an average PCC of **0.585** across OOD datasets, with all datasets showing positive correlation. This confirms that uncertainty signals contain meaningful information about prediction reliability.

## H.3. Interpretation

The combination of (1) elevated uncertainty on OOD datasets, (2) high AUROC, and (3) positive PCC demonstrates that RUAC exhibits *domain-level cautiousness*: the model correctly identifies domain shift and adopts a conservative stance, while still maintaining the ability to distinguish reliable from unreliable predictions within the high-uncertainty regime. This is precisely the behavior required for safe deployment in open-world scenarios.

# I. Risk-Coverage Analysis

To provide decision-relevant evidence that calibration improves independently of accuracy, we analyze the *risk-coverage* trade-off. This analysis addresses the concern that PAvPU improvements may be driven by accuracy gains rather than genuine calibration.

## I.1. Area Under Risk-Coverage Curve (AURC)

The risk-coverage curve plots segmentation risk (1 - accuracy) against coverage (fraction of predictions retained) as we progressively reject high-uncertainty predictions. A well-calibrated model should achieve lower risk at any given coverage level. The Area Under the Risk-Coverage Curve (AURC) provides a single scalar summary: **lower AURC indicates better uncertainty quality**.

Table S7 compares AURC between RUAC and Bayes-SAM2 (UE without adversarial training) across all 23 OOD datasets. RUAC achieves a mean AURC of **0.092** compared to 0.102 for Bayes-SAM2, representing a **9.8%** relative improvement. Crucially, this improvement is observed on **19/23 domains** (83%), demonstrating that the calibration benefit is consistent across diverse domain shifts.

*Table S7.* AURC comparison (lower is better). RUAC achieves lower AURC on 19/23 domains.

| Method | Mean AURC ↓ | Std |
|---|---|---|
| Bayes-SAM2 (UE) | 0.102 | 0.099 |
| **RUAC** | **0.092** | 0.093 |

## I.2. Selective Prediction Curves

Figure S8 shows selective prediction curves for representative domains. At each coverage level, we retain only the predictions with lowest uncertainty and measure the resulting accuracy. A model with better calibration achieves higher accuracy at any given coverage.

Across domains, RUAC (solid green) consistently lies above or matches Bayes-SAM2 (dashed blue), confirming that adversarial training improves the informativeness of uncertainty estimates for selective prediction. The improvement

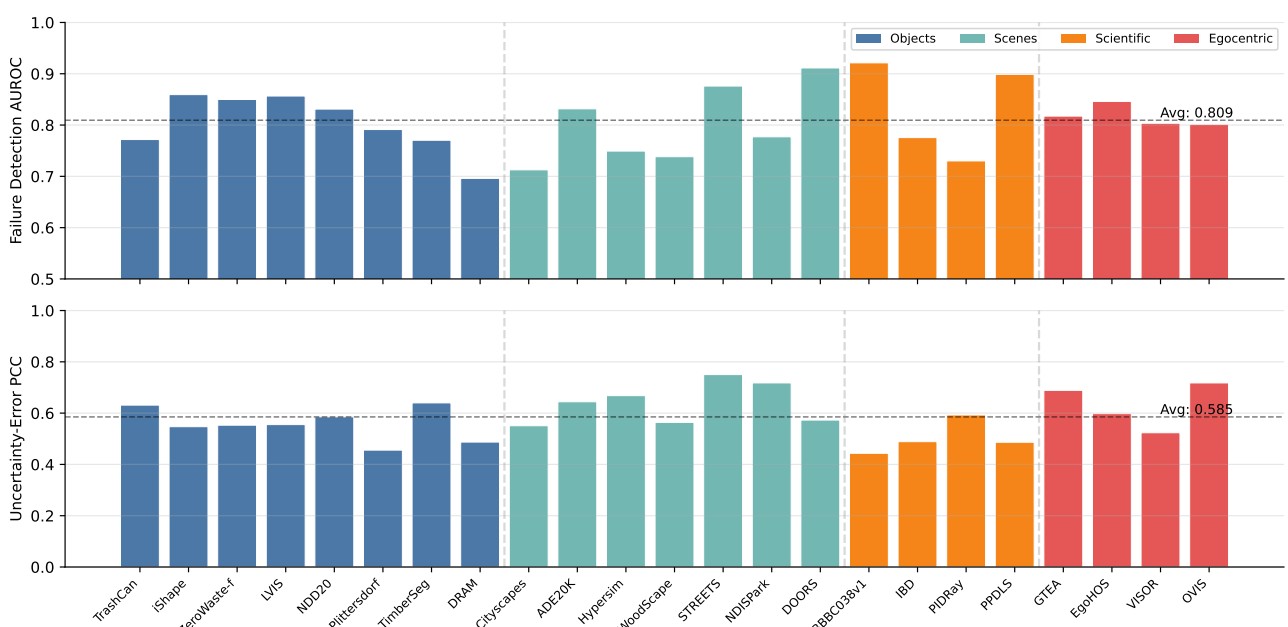

*Figure S6.* **Threshold-free uncertainty evaluation (pixel-level).** (Top) Failure Detection $\text{AUROC}_{pixel}$ measures whether pixel-wise uncertainty correctly ranks predictions by per-pixel accuracy (avg. 0.809). (Bottom) Uncertainty-Accuracy PCC confirms positive correlation. Both metrics evaluate at pixel granularity, i.e., each pixel is a sample with its own uncertainty and correctness label.

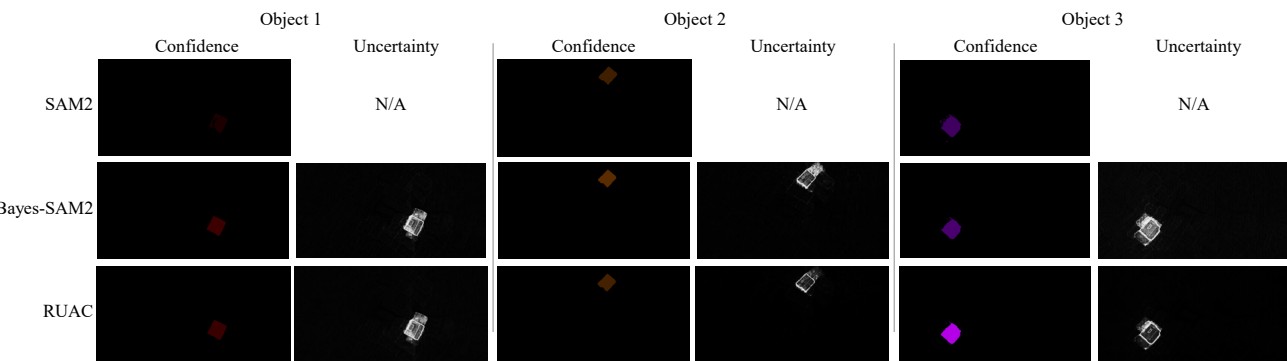

*Figure S7.* **Additional confidence and uncertainty visualization.** Complementing Figure 4 in the main text, we show three more object instances from additional rows of Figure 3. The layout follows the same format: each row corresponds to a method (SAM2, Bayes-SAM2, RUAC), displaying confidence and uncertainty maps. Brighter regions indicate higher confidence or uncertainty values. Consistent with the main text observations, RUAC demonstrates sharper confidence on object interiors while concentrating uncertainty along ambiguous boundaries, confirming its superior calibration behavior across diverse object instances.

is particularly pronounced on challenging domains (e.g., DRAM, PIDRay) where domain shift is severe.

### I.3. Implications for Decision-Making

These results demonstrate that RUAC's calibration improvement is *orthogonal to accuracy gains*:

1. **At matched accuracy**: When comparing predictions at the same coverage level (i.e., same effective accuracy), RUAC's uncertainty ranking is more reliable.

2. **Actionable uncertainty**: The improvement in AURC

directly translates to better performance in downstream tasks that use uncertainty for rejection, active learning, or human-in-the-loop correction.

3. **Consistent across domains**: The benefit is observed on the majority of OOD datasets, not just those where RUAC has higher accuracy.

## J. Statistical Significance Analysis

To rigorously evaluate calibration improvements, we conduct Wilcoxon signed-rank tests across all 23 OOD datasets. This non-parametric test is appropriate for paired compar-

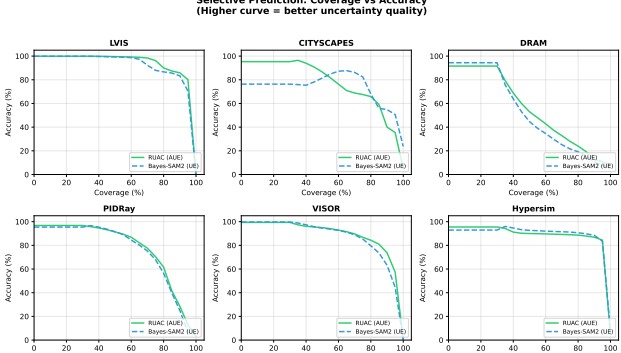

*Figure S8.* **Selective prediction curves** (Coverage vs. Accuracy). At each coverage level, we retain predictions with lowest uncertainty. Higher curves indicate better uncertainty quality. RUAC (solid) consistently achieves higher accuracy than Bayes-SAM2 (dashed) at matched coverage levels.

isons without assuming normality.

### J.1. Wilcoxon Signed-Rank Tests

Table S8 reports paired comparisons between RUAC and Bayes-SAM2 (UE only). For each metric, we count the number of domains where RUAC outperforms and compute the one-sided p-value.

*Table S8.* Wilcoxon signed-rank tests on calibration metrics across 23 OOD datasets. PAvPU denotes the threshold-averaged PAvPU over $\tau \in \{0.01, 0.05, 0.1\}$ as defined in Sec. 4. $*p < 0.05$, $**p < 0.01$, $***p < 0.001$.

| Metric | RUAC | Bayes-SAM2 | Wins | p-value |
|---|---|---|---|---|
| AURC $\downarrow$ | **0.092** | 0.102 | 19/23 | 0.0006*** |
| PAvPU $\uparrow$ | **63.7** | 55.4 | 19/23 | 0.0022** |
| ECE $\downarrow$ | **0.117** | 0.123 | 17/23 | 0.037* |
| AUROC$_{\text{mask}}$ $\uparrow$ | 0.971 | 0.966 | 13/23 | 0.22 |

### J.2. Interpretation

Three of four calibration metrics show statistically significant improvements:

- **AURC** ($p = 0.0006$): RUAC achieves lower risk at matched coverage levels on 83% of domains.
- **PAvPU** ($p = 0.0022$): Uncertainty-accuracy alignment improves significantly.
- **ECE** ($p = 0.037$): Expected calibration error decreases on 74% of domains.

The non-significant AUROC$_{\text{mask}}$ result is expected: AUROC$_{\text{mask}}$ (computed at mask granularity, averaging uncertainty per mask and comparing against mask-level IoU) measures *ranking* quality across entire masks, while our method optimizes *pixel-level calibration*. Both methods

achieve high AUROC$_{\text{mask}}$ ($>0.96$) because Bayesian uncertainty already provides excellent mask-level ranking. RUAC's contribution is improving *pixel-level calibration* (AUROC$_{\text{pixel}} = 0.81$, AURC, PAvPU).

## K. Channel Alignment Analysis

Unlike classification, where each image carries one label, segmentation requires per-pixel labels. This makes generic augmentation strategies fundamentally limited for our setting. Non-adversarial pixel perturbations cannot specifically discover the inputs on which the current decoder is suboptimal. Diffusion-based image synthesis risks semantic layout drift, breaking the pixel-mask correspondence required for supervision. Feature-statistics and style-transfer methods perturb images alone but cannot perturb their associated masks. RUAC sidesteps these limitations by jointly perturbing pixels and their spatial layout, enabling co-perturbation of images and ground-truth masks during training.

To understand why RUAC also outperforms these baselines empirically on both segmentation accuracy and calibration, we analyze how each method's training-time perturbation aligns with real OOD domain shift at the per-channel level of the mask decoder features.

### K.1. Method-Level Channel Alignment

For each augmentation method we extract 32-dimensional foreground pixel features (pooled within dilated ground-truth masks at stride-4 resolution) from the mask decoder and compute two quantities:

- **Augmentation shift**: the per-channel mean difference between augmented and clean features on the in-domain dataset (MOSE).
- **OOD shift**: the per-channel mean difference between OOD and in-domain features, averaged over the 23 zero-shot datasets.

**Channel alignment** is defined as the Pearson correlation across the 32 channels between these two shift vectors. A high correlation indicates that the augmentation method perturbs the same channels that actually shift under real domain change.

Figure S9 and Table S9 report the alignment correlation $r$ and the perturbation norm $\|\text{shift}\|$ for five representative methods. PGD and Patch are pixel-space attacks that do not directly perturb decoder features and are omitted from this feature-level analysis. The methods cluster into two regimes:

- **Low-alignment perturbations.** Random Noise, by construction, perturbs all channels uniformly and produces

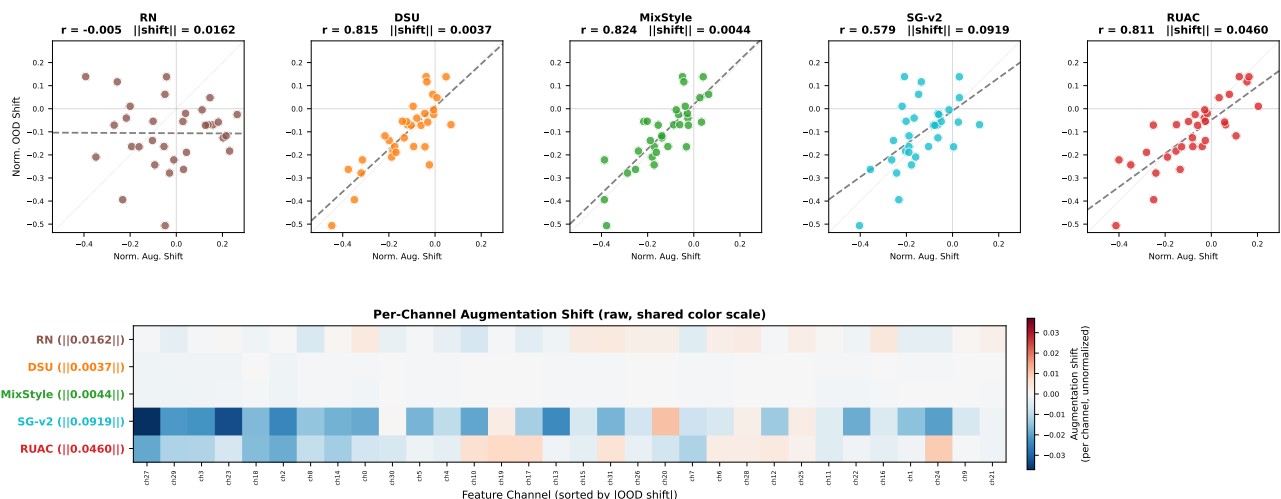

*Figure S9.* **Channel alignment across augmentation methods.** (Top) Scatter of per-channel augmentation shift (x-axis) versus per-channel OOD shift (y-axis), unit-normalized for visualization. The dashed line is the regression fit. (Bottom) Per-channel augmentation shift on the raw magnitude scale, sorted by aggregate OOD shift. Random Noise produces uncorrelated perturbations. Feature-statistics methods (DSU, MixStyle) achieve high alignment but very small magnitude. StyleGen-Targeted reaches moderate alignment at large magnitude. RUAC simultaneously achieves high alignment and substantial magnitude.

*Table S9.* **Channel alignment statistics for five augmentation methods.** Higher $r$ indicates better alignment with real OOD shift directions. $\|\text{shift}\|$ reflects perturbation magnitude on the same scale across methods. Effective robustness training requires both. RUAC is the only method that achieves high alignment together with substantial magnitude, without any target-domain access.

| Method | Type | $r$ | $\|\text{shift}\|$ |
|---|---|---|---|
| Random Noise | Pixel noise | $-0.005$ | 0.0162 |
| DSU | Feat. stats (Gaussian) | 0.815 | 0.0037 |
| MixStyle | Feat. stats (mixing) | 0.824 | 0.0044 |
| StyleGen-Targeted | SD style transfer | 0.579 | 0.0919 |
| **RUAC** | **Adv. style+deform** | **0.811** | **0.0460** |

near-zero correlation with real OOD shift. StyleGen-Targeted reaches moderate alignment (0.58) at large magnitude, but its diffusion-based image synthesis introduces semantic drift that hurts both J&F and PAvPU (Table 3).

- **High-alignment but low-magnitude perturbations.** DSU and MixStyle achieve the highest correlations ($\sim$0.82) because they perturb feature statistics in directions that match real OOD shift. However, their perturbation magnitude is roughly an order of magnitude smaller than RUAC's, so they cannot drive the decoder into the regions of feature space that robust fine-tuning needs to cover.

RUAC is the only method that satisfies both conditions. It learns adversarial style and deformation perturbations that align with the channels most affected by domain

shift ($r$=0.81) and applies them at substantial magnitude ($\|\text{shift}\|$=0.046), without any access to target-domain data.

### K.2. Per-Dataset Top-Channel Analysis

To localize the channels that actually shift under domain change, we compute the per-channel OOD shift for each of the 23 datasets, separately for the pretrained SAM2 encoder/decoder and for the fine-tuned Bayes-SAM2 model. Figure S10 shows the resulting heatmaps together with RUAC's learned adversarial perturbation magnitude per channel.

Three observations emerge from this analysis:

- **A few channels dominate OOD shift.** In the fine-tuned model (Figure S10c), channel ch27 appears in the top-3 most shifted channels for 18 of the 23 datasets, followed by ch29 (12 datasets) and ch17 (9 datasets). RUAC's adversarial perturbation (Figure S10a) targets exactly these channels. ch27 receives the largest adversarial shift among all 32 channels, even though RUAC receives no channel-level supervision.

- **Domain-specific channel patterns.** Different datasets engage different channels. PIDRay shifts most along ch27, ch29, and ch23, while DRAM shifts along ch10, ch17, and ch13. Visually similar domains share patterns. Driving scenes (Cityscapes, STREETS) both rely on ch27 and ch29. Indoor scenes (DOORS, DRAM) share ch10 and ch13.

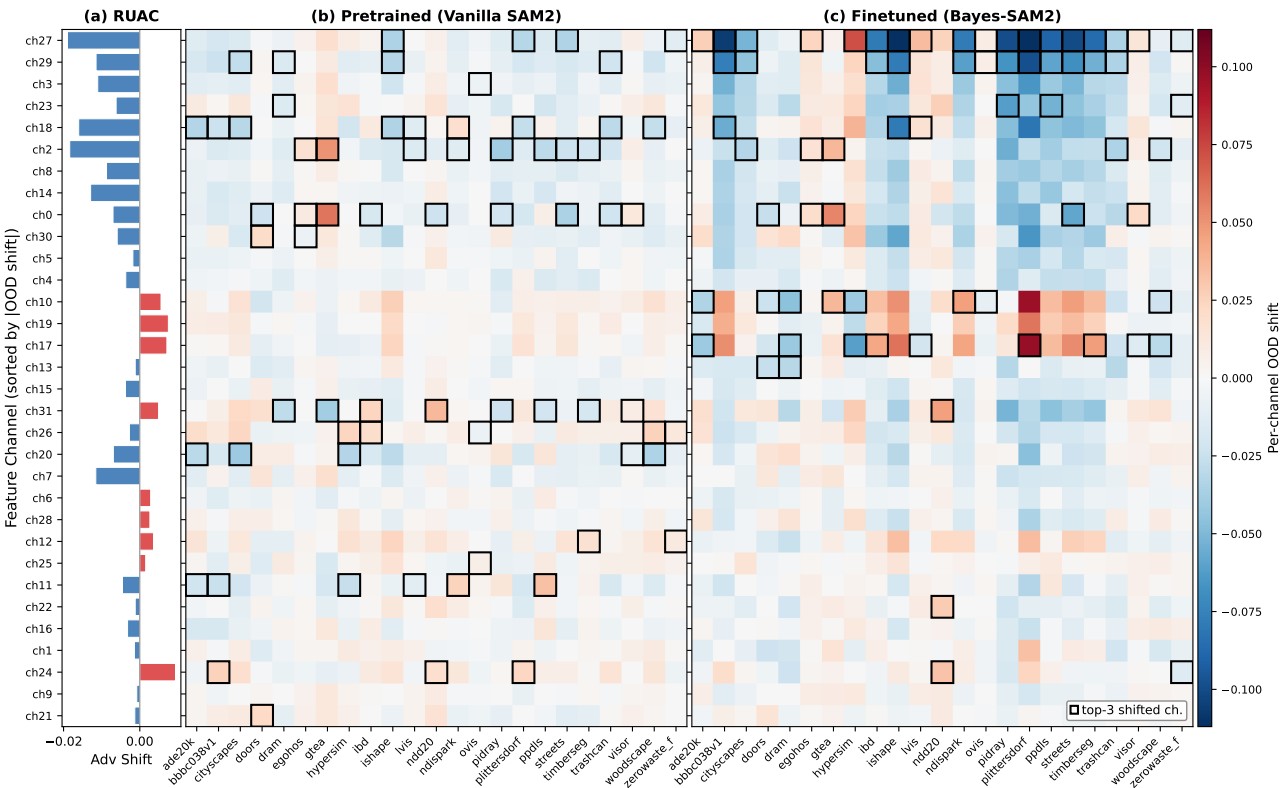

*Figure S10.* **Per-dataset channel-level OOD shift versus RUAC's learned perturbation.** (a) RUAC's adversarial perturbation magnitude per channel, learned end-to-end with no channel-level supervision. (b) Per-channel OOD shift heatmap for the pretrained SAM2 model. (c) Same heatmap for the fine-tuned Bayes-SAM2 model. Black boxes mark each dataset's top-3 most shifted channels. Channels are sorted by aggregate OOD shift magnitude (top = most shifted). Panels (b) and (c) share a color scale.

- **Fine-tuning concentrates the OOD shift.** Comparing the pretrained heatmap (b) with the fine-tuned heatmap (c) under the same color scale, fine-tuning amplifies the per-channel OOD shift. Each channel encodes more domain-specific information after fine-tuning, and the identity of the top-3 most shifted channels per dataset changes between (b) and (c). RUAC learns to attack the post-fine-tuning vulnerable channels, again without any channel-level supervision.

Together these observations explain the calibration gap in Table 3. Only RUAC perturbs the right feature directions at sufficient magnitude, and only RUAC discovers these directions purely from the adversarial training signal rather than from target-domain access.

## L. Hyperparameter Sensitivity Analysis

To assess RUAC's robustness to hyperparameter choices, we sweep the two main loss weights, $\beta$ (KL regularization, controlling the Bayesian decoder's posterior) and $\gamma$ (adversarial loss weight, controlling the contribution of the AUE branch), each across a $0.5\times$–$16\times$ range around the default. To keep

the sweep tractable we use a compressed 8-epoch schedule rather than the 20-epoch schedule used for the main results in Tables 1, 2, and 3. Absolute numbers therefore differ from the main results, but relative trends across the sweep range remain meaningful, since all sweep runs share the same training budget.

*Table S10.* **Sensitivity to the KL regularization weight $\beta$.** $\gamma$ is fixed at 0.2 (default). All metrics aggregated over 23 OOD datasets. The default configuration is highlighted.

| $\beta$ | J&F↑ | PAvPU↑ | AURC↓ | ECE↓ |
|---|---|---|---|---|
| 0.025 (0.5×) | 77.83 | 43.35 | 0.108 | 0.185 |
| 0.05 (1×, default) | 77.47 | 43.43 | 0.111 | 0.179 |
| 0.1 (2×) | 77.84 | 43.54 | 0.105 | 0.179 |
| 0.4 (8×) | 77.97 | 41.01 | 0.111 | 0.185 |
| 0.8 (16×) | 77.86 | 44.40 | 0.104 | 0.183 |

Across the full $0.5\times$–$16\times$ sweep for each hyperparameter, all four metrics remain stable: J&F varies by at most 0.7 points, AURC by $\leq 0.008$, PAvPU by $\leq 3.4$ points, and ECE by $\leq 0.014$. The sweep also reveals that the default $(\beta, \gamma) = (0.05, 0.2)$ is not necessarily optimal on the compressed schedule: a slightly larger $\gamma = 0.5$ improves J&F by

*Table S11.* **Sensitivity to the adversarial loss weight** $\gamma$. $\beta$ is fixed at 0.05 (default). All metrics aggregated over 23 OOD datasets. The default configuration is highlighted.

| $\gamma$ | J&F↑ | PAvPU↑ | AURC↓ | ECE↓ |
|---|---|---|---|---|
| 0.1 (0.5×) | 77.72 | 42.77 | 0.111 | 0.174 |
| 0.2 (1×, default) | 77.84 | 42.62 | 0.107 | 0.178 |
| 0.5 (2.5×) | 78.41 | 43.34 | 0.105 | 0.176 |
| 1.6 (8×) | 78.03 | 43.29 | 0.108 | 0.188 |
| 3.2 (16×) | 78.22 | 41.28 | 0.109 | 0.183 |

0.57 and AURC by 0.002 in Table S11. We did not perform a fine-grained search on the full 20-epoch schedule. We retain the original $(\beta, \gamma) = (0.05, 0.2)$ for consistency with all other results in the paper, treating the sensitivity sweep as evidence of robustness rather than an optimization target. The fact that RUAC outperforms all baselines (Tables 1, 2, 3) under a non-optimized configuration further supports that the gains stem from the AUE perturbation design rather than careful tuning.

## M. Loss Component Ablation

The full RUAC training objective combines the base segmentation loss (focal + dice) with two trainable components: the KL regularization on the Bayesian posterior (weighted by $\beta$) and the AUE adversarial loss (weighted by $\gamma$). We isolate the contribution of each by disabling it while keeping the rest unchanged. The base segmentation loss cannot be ablated, since training collapses without it. The combination $\beta = 0$ with $\gamma > 0$ is also not meaningful: the AUE adversarial loss relies on the Bayesian decoder's uncertainty map for calibration, so removing the KL regularization that shapes that posterior would leave the adversarial signal without a target. We therefore ablate the AUE adversarial loss ($\gamma=0$, equivalent to the no-augmentation Bayes-SAM2 in Table 1) and the KL regularization in turn. Results are aggregated over the 23 OOD datasets (Table S12).

*Table S12.* **Loss component ablation.** Aggregate metrics over 23 OOD datasets. The full RUAC configuration is highlighted, with **bold** marking the best value per metric.

| Configuration | J&F↑ | PAvPU↑ | AURC↓ | ECE↓ |
|---|---|---|---|---|
| RUAC (full) | **81.62** | **63.72** | 0.092 | 0.117 |
| w/o KL | 81.26 | 62.43 | **0.091** | **0.114** |
| w/o AUE | 79.87 | 55.40 | 0.102 | 0.123 |

Removing the AUE adversarial loss recovers the no-augmentation Bayes-SAM2 baseline and causes the largest drop ($\Delta$J&F $-1.75$, $\Delta$PAvPU $-8.32$, $\Delta$AURC $+0.010$, $\Delta$ECE $+0.006$), confirming that adversarial training is the primary contributor to RUAC's gains on both segmentation accuracy and uncertainty calibration. Removing the KL regularization causes only a minor change ($\Delta$J&F $-0.36$,

$\Delta$PAvPU $-1.29$, $\Delta$AURC $-0.001$, $\Delta$ECE $-0.003$). The differences in AURC and ECE between the full RUAC and the no-KL variant fall within run-to-run noise. The KL term primarily serves as a training stabilizer that prevents the Weibull posterior from degenerating, rather than as a performance driver. Both observations match our intuition: the adversarial branch carries the burden of aligning uncertainty with error under domain shift, while the KL regularization keeps the Bayesian decoder's posterior well-defined throughout training.

## N. Downstream Utility: Uncertainty-Guided Mask Correction

Beyond reporting calibration metrics, we test whether RUAC's uncertainty maps carry actionable signal for downstream decision-making. Inspired by SeCoV2 (Zhao et al., 2025), we apply a simple uncertainty-guided connected-component (CC) filter to each predicted mask: we extract connected components and remove fragments whose mean uncertainty exceeds an adaptive threshold, while always preserving the largest CC. The threshold is the 95th percentile of the predicted foreground uncertainty with a floor of 0.3 to prevent over-filtering on confident predictions. We refer to this post-processing step as UncCorr, and we apply it identically to Bayes-SAM2 and RUAC. Crucially, the same mask decoder and the same prompts are used. Only the uncertainty source differs.

Numerical results are reported in Table 4 of the main text. The two methods react oppositely to the same correction. UncCorr applied to Bayes-SAM2 *degrades* J&F on average ($-0.28$, hurting 14 of 23 datasets), confirming our calibration analysis: uncertainty that does not track error actively misleads downstream filters, sometimes removing correctly predicted regions. UncCorr applied to RUAC *improves* J&F (17 of 23 datasets), demonstrating that RUAC's uncertainty is well-aligned enough with error to identify and remove genuinely unreliable mask fragments. The +0.09 average gain is modest because UncCorr is a conservative post-processing step that only edits at the connected-component level, but the directional contrast (RUAC helps, Bayes-SAM2 hurts) is the substantive finding: better calibration translates directly into more useful uncertainty maps for downstream applications, not just better calibration scores. We expect more aggressive uncertainty-aware post-processing (e.g., per-pixel rejection, active sampling, or selective re-prompting) to amplify this gap.

## O. MC Sample Analysis

At inference time we estimate per-pixel uncertainty by drawing $S$ samples from the Weibull posterior of the Bayesian decoder. The forward pass that produces the predicted logits

uses the analytic Weibull expectation, so the segmentation output (J&F) is independent of $S$ by construction. The sample count only affects the uncertainty map quality. We sweep $S \in \{1, 5, 20, 50, 100\}$ on the same trained checkpoint and report aggregate metrics across the 23 OOD datasets in Table S13.

*Table S13.* **MC sample count $S$ at inference.** Same trained checkpoint, only the sample count is varied. The default $S = 20$ used in the main results is highlighted, with **bold** marking the best value per metric. Differences across $S$ are within run-to-run noise.

| $S$ | J&F↑ | PAvPU↑ | AURC↓ | ECE↓ |
|---|---|---|---|---|
| 1 | 81.53 | 63.36 | 0.096 | **0.115** |
| 5 | 81.44 | 63.32 | **0.089** | 0.116 |
| 20 (default) | **81.62** | 63.72 | 0.092 | 0.117 |
| 50 | 81.46 | **64.14** | 0.091 | 0.119 |
| 100 | 81.59 | 63.87 | 0.090 | 0.117 |

All four metrics are remarkably stable across the $100\times$ range of $S$. Even with a single sample ($S{=}1$), PAvPU is within 0.36 points and AURC within 0.007 of the default $S{=}20$. AURC reaches its best value already at $S{=}5$, indicating that the Weibull-based uncertainty is highly sample-efficient and saturates well below typical MC dropout sample counts. The minor J&F fluctuations ($\pm 0.21$) come from numerical precision since J&F is theoretically constant given the analytic forward pass. We therefore use $S{=}20$ as a conservative default that gives smooth uncertainty maps without meaningful additional cost. Users with tight latency budgets can drop to $S \in \{3, 5\}$ with negligible loss.

## P. Training Cost, Parameter Efficiency, and Extension

We report the practical training cost of RUAC, the parameter overhead introduced beyond the frozen SAM2 encoder, and how the design extends to future SAM models and to video tracking.

### P.1. Training Cost and Parameter Overhead

Training RUAC takes approximately 6 hours on 8 NVIDIA A40 GPUs (about 48 GPU-hours) for the standard 20-epoch schedule on the MOSE source domain. The trainable RUAC additions total approximately 0.45M parameters, which is 0.64% of SAM2's frozen image encoder (Hiera-B+, the hierarchical ViT image backbone of SAM2, 69.1M parameters). Table S14 reports the per-component breakdown: a compact deformation offset module ($\sim 370$K) dominates the budget, while the style attacker, GCN refinement module, and Bayesian pixel head are each on the order of 34K parameters or below. The deformation attacker additionally reuses a frozen copy of SAM2's pretrained memory encoder ($\sim 1.37$M), copied at initialization and kept frozen

throughout training to avoid GRL-induced gradient instability. These reused weights are not retrained and are excluded from the trainable count. The frozen image encoder is shared with the SAM2 baseline and is also not retrained.

*Table S14.* **Per-component trainable parameter breakdown of RUAC.** The SAM2 Hiera-B+ encoder (69.1M) is kept frozen. The deformation attacker additionally reuses a frozen copy of SAM2's memory encoder weights ($\sim 1.37$M, not shown) which is excluded from the trainable count.

| Component | Trainable Params |
|---|---|
| Style attacker (AdaIN + GRL) | $\sim 34$K |
| GCN refinement | $\sim 34$K |
| Deformation offset module | $\sim 370$K |
| Bayesian pixel head | $\sim 7.5$K |
| **Total trainable RUAC additions** | $\sim 0.45$M |
| Frozen SAM2 encoder | 69.1M |
| **Trainable overhead vs. encoder** | **0.64%** |

### P.2. Comparison with Diffusion-Based Augmentation

Recent style-augmentation pipelines often rely on pretrained diffusion models to synthesize domain-shifted training images (Dunlap et al., 2023; Trabucco et al., 2024). We list the parameter scale of representative diffusion backbones in Table S15 for reference. RUAC introduces roughly three orders of magnitude fewer trainable parameters than these alternatives. Because RUAC perturbs the segmentation feature space directly rather than synthesizing pixels, it also avoids the semantic drift, prompt-engineering cost, and inference-time text-encoder dependency typical of diffusion-based augmentation.

We instantiate diffusion-based augmentation as two variants of **StyleGen**, both using SD 1.5 img2img following ALIA (Dunlap et al., 2023) and DA-Fusion (Trabucco et al., 2024). **StyleGen-Generic** uses 2 prompts spanning all target domains. **StyleGen-Targeted** uses 9 prompts, each tailored to one of the 11 OOD target domains below the average J&F (overlapping domains merged, e.g., VISOR+GTEA into "egocentric kitchen"), representing a strong prompt-engineering effort that explicitly targets the test distribution. Despite this, StyleGen-Targeted *degrades* below StyleGen-Generic and even below Random Noise on J&F and PAvPU (Table 3): more aggressive prompt engineering toward OOD domains amplifies semantic drift in the synthesized images, breaking the pixel-mask correspondence required for segmentation supervision. This is the empirical evidence behind the channel-alignment analysis in Appendix K.1, where StyleGen-Targeted achieves only moderate alignment ($r{=}0.58$) at large magnitude.

*Table S15.* **Parameter scale of diffusion-based alternatives vs. RUAC.** Counts refer to the augmentation backbone as reported in the cited papers.

| Method | Parameters |
|---|---|
| Stable Diffusion 1.5 (Rombach et al., 2022) | $\sim 860$M |
| SDXL (Podell et al., 2024) | $\sim 2.6$B |
| ControlNet (Zhang et al., 2023) | $\sim 361$M |
| InstructPix2Pix (Brooks et al., 2023) | $\sim 860$M |
| **RUAC (ours)** | $\sim \mathbf{0.45}$**M** |

## P.3. Extension to Future SAM Models and Video

The RUAC trainable modules attach to the mask decoder and operate on the contrastive feature space produced by the frozen Hiera-B+ trunk. The encoder itself is untouched during training. This decoder-attached, encoder-agnostic design means that swapping the encoder for a future model, e.g., the SAM3 image encoder (Carion et al., 2025) or a video tracker backbone, only requires re-targeting the decoder feed: the adversarial style and deformation networks, the GCN refinement module, and the Bayesian mask decoder can be reused without architectural change. Because the encoder remains frozen, the cost of porting RUAC to a new backbone is dominated by re-running the same $\sim 48$ GPU-hour fine-tuning of the $\sim 0.45$M trainable RUAC parameters, not by retraining the much larger encoder. For video tracking specifically, the Bayesian decoder's per-pixel uncertainty can be temporally aggregated to flag tracker drift across frames. We leave this empirical study to future work.

