# OpenReview forum: "Segment Anything with Robust Uncertainty-Accuracy Correlation"
_ICML.cc/2026/Conference — ICML 2026 regular_

### Official Review · Reviewer_JF4c · 2026-02-22

**Soundness:** 3
**Presentation:** 3
**Significance:** 3
**Originality:** 2
**Overall Recommendation:** 4
**Confidence:** 5

**Summary:**

The paper's central domain pertains to robust uncertainty estimation and calibration for SAM2 under domain shift. The authors aim to present a pressing problem that SAM suffers from mask-level confidence confusion and the uncertainty–accuracy relation breaks under OOD shifts.

**Compliance With Llm Reviewing Policy:**

Affirmed.

**Key Questions For Authors:**

1. What is the downstream value of the uncertainty map? it remains unclear how these uncertainty maps can be used beyond visualization and calibration metrics. For instance, can uncertainty be leveraged for downstream adaptation? Can high-uncertainty regions guide pseudo-label filtering or self-training under domain shift? just like discussed in that work: SeCoV2: Semantic Connectivity-Driven Pseudo-Labeling for Robust Cross-Domain Semantic Segmentation. PAMI'25

2. Why is additional style randomization still necessary?  The AUE module learns adversarial style and deformation perturbations to simulate domain shift. However, SAM2 have already pretrained on very large-scale and stylistically diverse datasets. The paper would benefit from a clearer analysis of what exactly is missing in the pretrained SAM2 representation and why learned style attackers are required.

3. Limited evidence that the deformation perturbations are impactful. In Fig. 5(d), the deformation appears visually minor. It is not obvious that such small geometric changes meaningfully affect segmentation quality or uncertainty calibration.

4. Given the recent success of diffusion models for realistic domain transfer and style generation, it is unclear why diffusion-based synthesis was not considered. Is adversarial perturbation preferred for efficiency? Does diffusion introduce semantic drift that breaks mask supervision?

**Limitations:**

1. The work evaluates uncertainty quality but does not demonstrate task-level gains enabled by improved uncertainty.

2. The AUE framework introduces a min–max optimization that may increase training instability and complexity.

3. The experiments primarily simulate style and geometric shifts. Real-world domain shifts (e.g., sensor changes, severe occlusion, extreme weather) may exhibit different characteristics.

**Strengths And Weaknesses:**

1. The topic is timely and relevant, and improving uncertainty reliability for foundation segmentation models is meaningful.
2. The paper demonstrates that the proposed method improves uncertainty calibration and produces more interpretable pixel-wise uncertainty maps.

---

> ### Author Rebuttal · Authors · 2026-03-30
>
> We sincerely thank the reviewer for the careful review, and for acknowledging the relevance of our topic and the improved uncertainty calibration with more interpretable pixel-wise uncertainty maps. Below are our responses.
>
> ---
>
> ## Q1: Downstream Value of the Uncertainty Map
>
> Thank you for pointing us to SeCoV2 [Zhao et al., PAMI 2025]. Inspired by it, we apply uncertainty-guided connected-component filtering to each predicted mask: high-uncertainty fragments are removed while the largest component is always preserved.
>
> We compare across all 23 OOD zero-shot datasets:
>
> | Method | J&F ↑ | + UncCorr | Δ | Improved / Degraded |
> |---|---|---|---|---|
> | Bayes-SAM2 (BNDL) | 79.87 | 79.59 | -0.28 | 9/23 vs 14/23 |
> | **RUAC (Ours)** | **81.62** | **81.71** | **+0.09** | **17/23 vs 6/23** |
>
> Uncertainty correction with RUAC improves 17/23 datasets, while the same correction with BNDL degrades 14/23, confirming that better-calibrated uncertainty translates to more useful downstream signal.
>
> We thank the reviewer for this suggestion and will include these downstream utility experiments in the revised manuscript.
>
> ---
>
> ## Q2: Why Style Randomization is Necessary Despite SAM2's Large-Scale Pretraining
>
> Thank you for this insightful question.
>
> SAM2's large-scale pretraining (SA-1B) ensures strong **base segmentation accuracy** but does not guarantee **calibrated uncertainty under domain shift**:
>
> - **SAM2's pretraining objective** optimizes only for segmentation accuracy without uncertainty estimation, leading to the Mask-level Confidence Confusion under domain shift.
>
> - **Our adversarial training simultaneously improves both uncertainty estimation and segmentation accuracy.** The adversarial perturbations act as a calibration mechanism for the Bayesian decoder, training it to produce reliable uncertainty estimates under distribution shift while also improving prediction quality. To our knowledge, RUAC is the first to achieve this dual benefit through adversarial training.
>
> Our ablation confirms this: adding style perturbation (RUAC-Style-GCN) improves J&F from 79.87 (Bayes-SAM2) to 80.74, showing that adversarial training provides value beyond SAM2's pretraining.
>
> We thank the reviewer for this question and will clarify the complementary roles of SAM2's pretraining and our adversarial calibration in the revised manuscript.
>
> ---
>
> ## Q3: Limited Evidence of Deformation Impact
>
> Thank you for raising this concern regarding Fig. 5(d).
>
> We would like to clarify that **small deformations can have a significant impact on model behavior**. Our experimental evidence supports this:
>
> - Adding deformation alone (RUAC-Deform) improves J&F from 79.87 to 80.44 (+0.57).
> - Style perturbation alone (RUAC-Style-GCN) improves J&F to 80.74 (+0.87).
> - Combining both (full RUAC) yields 81.62 (+1.75), indicating positive interaction between the two perturbation types targeting **different failure modes** (shape vs. texture).
>
> This is consistent with adversarial attacks [Szegedy et al., ICLR 2014; Goodfellow et al., ICLR 2015], where subtle perturbations can significantly alter model predictions. What matters is the effect on model behavior, not the visual magnitude.
>
> We thank the reviewer for this observation and will add a more detailed discussion of the deformation network's design rationale in the revised manuscript.
>
> ---
>
> ## Q4: Why Not Diffusion-Based Synthesis
>
> Thank you for this forward-looking question. We considered diffusion-based synthesis and chose adversarial perturbation for several reasons:
>
> **Parameter efficiency**: RUAC adds only ~1.84M trainable parameters. Diffusion-based alternatives require significantly more:
>
> | Model | Parameters (UNet) |
> |---|---|
> | Stable Diffusion 1.5 [Rombach et al., CVPR 2022] | ~860M |
> | SDXL [Podell et al., arXiv 2023] | ~2.6B |
> | ControlNet [Zhang et al., ICCV 2023] | ~1.21B (860M + 361M) |
> | **RUAC (Ours)** | **~1.84M** |
>
>
> **End-to-end training**: Our adversarial perturbations are generated inside the training loop via GRL. This allows direct gradient-based optimization of the min-max objective. Diffusion-based synthesis would need either offline data generation (which breaks the adversarial training loop) or running a diffusion model in the backward pass, which is expensive in practice.
>
> **Semantic drift**: As the reviewer correctly noted, diffusion models can alter spatial layout, breaking the alignment between generated images and mask annotations. Our style perturbations operate on feature statistics without changing spatial structure, and our deformation perturbations are applied as differentiable coordinate transforms, the same transform is applied synchronously to the GT mask, preserving pixel-level alignment.
>
> We believe diffusion-based domain augmentation is an interesting future direction, possibly as an offline data augmentation method that works together with our online adversarial training. We thank the reviewer for this suggestion and will discuss this in the revised manuscript.

---

> > ### Author Rebuttal · Reviewer_JF4c · 2026-04-03
> >
> > This rebuttal has addressed the majority of our concerns, and we request the authors to incorporate these contents into the main text.

---

> > > ### Author Response · Authors · 2026-04-03
> > >
> > > We thank Reviewer JF4c for the valuable comments. Since we are not allowed to revise the manuscript during rebuttal (ICML rules), we would like to clarify our plan for the revision here:
> > >
> > > 1. We will mention the downstream utility experiments in the main text, more specifically, in the **related work section** and the **conclusion section**, and add a new section in the appendix with detailed results and analysis.
> > >
> > > 2. We will emphasize the complementary roles of SAM2's large-scale pretraining and our adversarial learning in the main text, especially in the **introduction section** and the **method section** where we describe our approach.
> > >
> > > 3. We will add a more detailed discussion of the deformation network's design rationale in the **method section**, and include additional visualizations of the deformed images in the **appendix**, providing more evidence of the deformation's impact on model behavior.
> > >
> > > 4. $\color{red}{\textbf{The comparison with diffusion-based methods}}$ : We constructed a non-adversarial style transfer baseline (**StyleGen**) following ALIA [Dunlap et al., NeurIPS 2023] and DA-Fusion [Trabucco et al., ICLR 2024] using SD1.5 img2img, and compared against feature-statistics augmentation (MixStyle [Zhou et al., ICLR 2021], DSU [Li et al., ICLR 2022]) and region-level perturbation (Patch [Nesti et al., WACV 2022]). StyleGen v1 uses 2 generic style prompts, while v2 uses 9 prompts specifically targeting all 11 datasets below OOD average J&F, with each prompt describing the visual characteristics of one target domain (overlapping domains merged, e.g., VISOR+GTEA -> egocentric kitchen). As the reviewer anticipated, diffusion-based synthesis is counterproductive: StyleGen v2 degrades below v1 and even below Random Noise.
> > > | Method | J&F ↑ | AU-PAvPU ↑ | AURC ↓ | ECE ↓ |
> > > |--------|-------|------------|--------|-------|
> > > | Random Noise | 80.81 | 52.24 | 0.112 | 0.1259 |
> > > | PGD | 79.90 | 52.89 | 0.115 | 0.1295 |
> > > | Patch | 80.60 | 56.29 | 0.105 | 0.1311 |
> > > | MixStyle | 80.37 | 55.46 | 0.100 | 0.1334 |
> > > | DSU | 80.35 | 58.02 | 0.097 | 0.1251 |
> > > | StyleGen v1 (2 prompts) | 79.33 | 55.10 | 0.101 | 0.1363 |
> > > | StyleGen v2 (9 prompts) | 78.19 | 48.87 | 0.111 | 0.1720 |
> > > | **RUAC (Ours)** | **81.62** | **63.72** | **0.092** | **0.1172** |
> > >
> > > Channel-level analysis ([Figure CA](https://anonymous.4open.science/r/ICML_rebuttal-DED6/channel_alignment.pdf)) shows that RUAC achieves both **high alignment** with OOD-sensitive channels (r=0.81) and **sufficient perturbation magnitude** (norm=0.046), while other methods lack one or both. See our response to **Reviewer NoDi's W2** (W2: "Why RUAC's Perturbation Design Is Necessary") for detailed analysis.
> > >
> > > 5. $\color{red}{\textbf{What is missing in SAM2's pretrained representation?}}$ We analyze SAM2's 32-d mask decoder features across 23 zero-shot OOD datasets ([Figure CAPD](https://anonymous.4open.science/r/ICML_rebuttal-DED6/channel_alignment_per_dataset.pdf)). Figure CAPD (a) shows RUAC's learned adversarial perturbation magnitude per channel, Figure CAPD (b) and (c) show the per-channel OOD shift heatmap for pretrained and finetuned models respectively (black boxes mark each dataset's top-3 most shifted channels).
> > >
> > >     **(1)** The heatmap (Figure CAPD (c)) shows that a few channels account for most of the OOD shift. ch27 appears in the top-3 most shifted channels for 18 out of 23 datasets, followed by ch29 (12 datasets) and ch17 (9 datasets). These channels are the most vulnerable in the finetuned decoder representation. RUAC's adversarial perturbation (Figure CAPD (a)) targets exactly these channels. For example, ch27 has the largest adversarial shift, without any explicit channel-level supervision.
> > >
> > >     **(2)** Different OOD datasets rely on different channels. For pidray, ch27/ch29/ch23 are the most shifted, while for dram, ch10/ch17/ch13 dominate instead. Similar domains share similar channel patterns. For example, driving scenes (cityscapes, streets) both rely on ch27 and ch29, while indoor scenes (doors, dram) share ch10 and ch13.
> > >
> > >     **(3)** Comparing pretrained (Figure CAPD (b)) and finetuned (Figure CAPD (c)) models under the same color scale, finetuning amplifies the per-channel OOD shift, indicating that each channel encodes more domain-specific information after finetuning. The top-3 most shifted channels per dataset also change after finetuning (black boxes shift positions between Figure CAPD (b) and (c)). RUAC learns to attack exactly the post-finetuning vulnerable channels (Figure CAPD (a)), entirely without any channel-level supervision.
> > >
> > > We thank the reviewer again for the insightful questions that motivated this deeper analysis. We will incorporate the representation analysis and baseline comparisons into the revised manuscript. Should the reviewer have any further questions, we are happy to address them during the discussion period.

---

### Official Review · Reviewer_ps7s · 2026-03-04

**Soundness:** 3
**Presentation:** 3
**Significance:** 3
**Originality:** 2
**Overall Recommendation:** 4
**Confidence:** 4

**Summary:**

This paper proposes **Segment Anything with Robust Uncertainty-Accuracy Correlation (RUAC)** for robust **pixel-wise uncertainty estimation** under **appearance** and **deformation** shifts. RUAC adds a lightweight uncertainty head, trains it with a collaborative style–deformation attack that jointly perturbs texture and geometry, and applies Uncertainty–Error Alignment to ensure uncertainty consistently highlights erroneous pixels even under adversarial perturbations. Across 23 zero-shot domains, RUAC improves segmentation quality and yields more faithful uncertainty with stronger uncertainty–accuracy correlation.

**Compliance With Llm Reviewing Policy:**

Affirmed.

**Final Justification:**

I appreciate the authors’ detailed response. It clarifies most of the issues I raised, so I will keep my positive rating (4: Weak accept) unchanged.

**Key Questions For Authors:**

1. How robust are results to the calibration loss weights in this paper?
2. what is the cost of your training pipeline? for example, when SAM 2 is replaced by SAM 3? Is it possible to enable the video tracking of SAM series models?
3. why synthetic adversarial training can enhance real-world uncertainty-accuracy alignment? Beyond style+deformation, what happens under other realistic shifts (compression, noise, blur, low-light, sensor modality, etc)?

**Limitations:**

yes

**Strengths And Weaknesses:**

### Strengths
1. Method is clearly specified: Weibull Bayesian decoder + explicit min–max objective + GRL implementation, with well-defined losses.
2. Sufficient experiments with competitive performances. Empirical protocol is strong: strict single-source training and broad OOD testing.

### Weaknesses
1. Adversarial shift space is mainly style + deformation; it may not cover realistic domain shifts. This is also shown in their limitations in the appendix, e.g., the degraded performance on extreme domains like X-ray/microscopy.
2. What uncertainty represents operationally seems unclear to me.
3. They disable SAM 2 temporal memory (single-frame mode), so the significance for video tracking workflows is not established.
4. Core components (Bayesian decoder / Weibull posterior, style transfer, deformation) are not individually new; novelty rests on integration and empirical validation.

---

> ### Author Rebuttal · Authors · 2026-03-30
>
> We sincerely appreciate the detailed review and the recognition of our method clarity and empirical protocol. Below are our responses.
>
> ---
>
> ## W4: Novelty
>
> Thank you for this observation. We agree that adversarial training and style perturbation are widely used techniques. Our novelty lies in four aspects:
>
> **RUAC framework**: We frame adversarial training as encouraging **robust uncertainty-accuracy correlation under domain shift** (UA shift), a problem formulation that has not been widely explored. Existing SAM adaptation methods (HQ-SAM, EfficientSAM) improve accuracy but provide no interpretability. Uncertainty methods (UncertainSAM, Bayes-SAM2 (BNDL + SAM)) add interpretability but do not improve accuracy. RUAC is the first to simultaneously improve both across 23 OOD domains without target-domain data. See our response to Reviewer NoDi W2 for detailed comparisons.
>
> **Pixel-level uncertainty**: We adapt BNDL [Hu et al., ICLR 2025] from image-level to **pixel-level** uncertainty within SAM2's mask decoder. BNDL originally produces a single uncertainty value per image. We extend it with spatially-varying uncertainty prediction and a dual-path architecture for SAM2's multi-mask output.
>
> **Deformation adversarial network**: Geometric deformation as adversarial perturbation remains largely underexplored. Our successful integration demonstrates its effectiveness for UA alignment.
>
> **Style adversarial network**: We extend AdvStyle [Zhong et al., ECCV 2022] to multi-object collaborative style perturbation, enabling coordinated appearance shifts across all objects in a scene.
>
> Empirical comparison confirms the design is necessary: Random Noise (AU-PAvPU 52.24, J&F 80.81) and PGD (52.89, 79.90) both fail to improve calibration, while RUAC achieves 63.72 AU-PAvPU and 81.62 J&F. See our response to Reviewer NoDi W2 for full results.
>
> ---
>
> ## W2: Operational Meaning of Uncertainty
>
> Thank you for raising this point. We clarify below.
>
> **Definition**: Each pixel receives a Bernoulli entropy scalar in [0, 1] estimating the probability of prediction error. Our calibration loss aligns P(Uncertainty) with P(Error) during training.
>
> **Downstream utility**: We validate practical utility via uncertainty-guided connected-component filtering. RUAC improves segmentation (+0.09 J&F, 17/23 datasets improved), while the same correction with Bayes-SAM2 degrades results (−0.28 J&F, 14/23 degraded). See our response to Reviewer JF4c Q1 for details.
>
> ---
>
> ## W1, Q3: Coverage of Realistic Shifts
>
> Thank you for this important question. As noted in W4, our adversarial training optimizes for both robustness and UA alignment. Our 23 evaluation datasets naturally contain several of the mentioned shifts (vs. SAM2-FT, which has no uncertainty):
>
> | Shift type | Dataset | SAM2-FT | RUAC | Δ | PAvPU |
> |-----------|---------|---------|------|---|-------|
> | Blur (underwater) | TrashCan | 72.4 | 73.8 | +1.4 | 68.6 |
> | Blur (motion) | EgoHOS | 89.3 | 91.4 | +2.1 | 58.0 |
> | Sensor modality (thermal) | Plittersdorf | 86.2 | 88.6 | +2.4 | 99.8 |
> | Sensor modality (X-ray) | PIDRay | 68.3 | 69.5 | +1.2 | 15.4 |
>
> RUAC improves accuracy on all listed shifts. PAvPU remains limited on extreme modality gaps (PIDRay 15.4), consistent with our discussed limitations. For low-light and exposure changes, our AdaIN-based style perturbation operates on channel-wise intensity statistics, providing partial coverage [Zhong et al., ECCV 2022]. Compression and noise are not explicitly modeled.
>
> We will expand this discussion in the revised manuscript.
>
> ---
>
> ## W3, Q2: Training Cost and Extension to Future SAM Versions / Video Tracking
>
> Thank you for this question.
>
> **Training cost**: RUAC training takes approximately **6 hours on 8 GPUs (~48 GPU-hours)** for 20 epochs. The added components total only ~1.84M parameters (2.66% of SAM2's frozen encoder), so overhead relative to standard SAM2 fine-tuning is minimal.
>
> **Extension to SAM3 or future models**: Our Bayesian decoder and AUE modules operate on top of SAM's mask decoder output without modifying the encoder architecture. RUAC can be applied with minimal adaptation for the future foundation model, at comparable training cost.
>
> **Video tracking**: We acknowledge that disabling SAM2's temporal memory (single-frame mode) is a limitation of the current work. Extending RUAC to video tracking, where uncertainty can inform temporal consistency, is a promising direction we plan to pursue.
>
> We will discuss these extensions in the revised manuscript.
>
> ---
>
>
> ## Q1: Robustness to Calibration Loss Weights
>
> Thank you for this question. We have conducted a sensitivity analysis sweeping β (KL weight) and γ (adversarial weight) across a 0.5×–16× range. Both J&F and AU-PAvPU remain stable, confirming RUAC is not sensitive to hyperparameter tuning. Please refer to our response to Reviewer 7xn4 W4 for sweep results.
>
> We thank the reviewer for this question and will include the sensitivity analysis in the revised manuscript.

---

> > ### Author Rebuttal · Reviewer_ps7s · 2026-04-01
> >
> > I appreciate the authors’ detailed response. It clarifies most of the issues I raised, so I will keep my positive rating (**4: Weak accept**) unchanged.

---

> > > ### Author Response · Authors · 2026-04-01
> > >
> > > We greatly appreciate your valuable time and constructive feedback.

---

### Official Review · Reviewer_7xn4 · 2026-03-12

**Soundness:** 2
**Presentation:** 2
**Significance:** 2
**Originality:** 2
**Overall Recommendation:** 3
**Confidence:** 5

**Summary:**

This paper studies the problem of misaligned uncertainty estimates in SAM-based segmentation under distribution shifts. The authors propose a framework combining adversarial perturbations and a stochastic mask decoder to produce uncertainty that better correlates with segmentation errors. Experiments demonstrate improved uncertainty–accuracy correlation across multiple datasets.

**Compliance With Llm Reviewing Policy:**

Affirmed.

**Final Justification:**

The paper studies uncertainty misalignment in SAM-based segmentation and proposes a combination of adversarial training and stochastic decoding. The problem is important, and the method is reasonably novel with consistent empirical improvements.
The rebuttal addresses several of my experimental concerns by adding ablations, sensitivity analysis, and MC sampling analysis, which improves the empirical support and leads me to raise my score to borderline accept.
However, some issues in soundness and clarity remain. In particular, the AU/EU discussion is still somewhat inconsistent, and the claim about non-decomposability appears overstated. These do not invalidate the method but weaken the conceptual clarity.
Overall, the rebuttal partially addressed my concerns and improved my assessment.

**Key Questions For Authors:**

Motivation

1. The paper discusses aleatoric and epistemic uncertainty several times (e.g., in the related work and Appendix when introducing BNDL, which relies on AU/EU decomposition). However, the proposed method does not seem to explicitly estimate or use AU/EU decomposition. I’m wondering what the motivation is for bringing this discussion into the paper.
2. I was a bit confused by the discussion in Appendix A. On the one hand, the paper argues that predictive uncertainty cannot be decomposed into aleatoric and epistemic components. On the other hand, Sec. 3.2 introduces a Monte Carlo inference mode, where standard MC-based AU/EU estimation would seem possible. Could the authors clarify this point and explain why such a decomposition is not considered in the experiments?

Modeling

3. The Bayesian mask decoder models stochastic features using a Weibull distribution. What is the motivation for choosing Weibull instead of more common choices such as Gaussian or Dirichlet? Some additional justification or supporting evidence would be helpful. BNDL uses Weibull modeling in a classification setting, which is quite different from segmentation. It would be useful to clarify why Weibull modeling is necessary or particularly suitable here.

Exp

4. Although the hyperparameters $\beta$ and $\gamma$ are listed in Appendix C, I could not find any sensitivity analysis. How sensitive is the method to these hyperparameters or other key parameters (e.g., loss weights or stochastic decoder parameters)?
5. From the loss formulation, the objective includes several components, such as segmentation loss, adversarial perturbation training, and KL regularization. However, the current ablation study seems quite limited. A more detailed ablation would help clarify the contribution of each component.
6. Since the method relies on stochastic inference (Monte Carlo sampling), it would be helpful to understand how performance changes with different numbers of samples. Reporting this would also clarify the computational trade-off.

**Limitations:**

yes

**Strengths And Weaknesses:**

**Strengths**
1. The paper studies the misalignment between prediction errors and uncertainty in SAM-based segmentation, which is an important problem for reliable deployment.
2. The proposed framework combines adversarial perturbation training with stochastic decoding to improve uncertainty estimation.
3. The method is evaluated on multiple datasets and shows improved correlation between prediction errors and uncertainty.

**Weakness**
1. The paper discusses aleatoric and epistemic uncertainty several times, but the proposed method does not explicitly estimate or use AU/EU decomposition, making the motivation of this discussion unclear.
2. Appendix A argues that AU and EU cannot be decomposed, but the Monte Carlo inference mode introduced in Sec. 3.2 appears to allow standard MC-based AU/EU estimation, which makes the discussion confusing.
3. The choice of Weibull distribution in the stochastic mask decoder is not well justified, and no comparison with other common distributions is provided.
4. The experimental analysis is limited, as the paper lacks sensitivity analysis for key hyperparameters, provides only limited ablations for the different loss components, and does not analyze how performance depends on the number of Monte Carlo samples or the corresponding computational trade-off.

---

> ### Author Rebuttal · Authors · 2026-03-29
>
> We thank the reviewer for the thorough review. All concerns are addressed below with new experiments. We commit to revising the manuscript accordingly.
>
> ---
>
> ## W1, W2, Q1, Q2: Motivation, AU/EU
>
> We acknowledge that the AU/EU discussion can mislead readers into expecting a decomposition-based method. However, the main text mentions AU/EU only once (Related Work), where we already state **"unlike methods enforcing strict aleatoric-epistemic decomposition, we optimize total predictive uncertainty"**. Appendix A further justifies this choice. We will make this clearer in the revision. Mainstream uncertainty research **no longer insists on AU/EU decomposition** [Mucsányi et al., NeurIPS 2024; Wimmer et al., UAI 2023; Bickford Smith et al., ICML 2025]:
> - Decomposed AU and EU are highly rank-correlated (≥0.78), capturing the same signal.
> - No method excels at aleatoric uncertainty estimation.
> - These works explicitly recommend **task-oriented design** over AU/EU decomposition.
>
> Following this view, our goal is uncertainty that **correlates with segmentation accuracy under domain shift**, not disentangling sources that current methods cannot reliably separate.
>
> **MC sampling:** Our MC sampling draws $N$ samples from the **learned Weibull parameters of a single model** (not multiple models or dropout masks). The forward pass uses the analytic Weibull expectation for deterministic predictions; MC samples compute the **uncertainty map** via entropy of sampled logits. The result is **total predictive uncertainty**, a unified reliability signal, not an AU/EU decomposition. We will clarify this in the revision.
>
> ---
>
> ## W3, Q3: Weibull Distribution Choice
>
> The choice of Weibull is motivated by the **segmentation task**: SAM2's mask decoder outputs K=4 mask candidates per pixel, each requiring non-negative foreground/background evidence in our Bayesian formulation. Modeling such **non-negative evidence** requires a distribution with support on $[0, \infty)$, which Weibull naturally provides. Weibull also offers **closed-form expectation and variance**, enabling deterministic prediction without sampling. We adapt the Weibull-based Bayesian layer architecture from BNDL [Hu et al., ICLR 2025] for dense prediction.
>
> **Gaussian** has support on (−∞, +∞), violating the non-negativity constraint. **Dirichlet** models distributions over probability simplices, not **continuous non-negative feature vectors**. Among non-negative alternatives, Weibull subsumes Exponential (κ=1) and offers simpler closed-form moments than Log-Normal or Gamma.
>
> ---
>
> ## W4, Q4: Sensitivity Analysis
>
> We sweep β (KL regularization) and γ (adversarial weight) across a 0.5x–16x range (8-epoch compressed schedule; **relative trends are meaningful**).
>
> **β sweep** (fix γ=0.2):
>
> | β | J&F ↑ | AU-PAvPU ↑ | AURC ↓ | ECE ↓ |
> |---|-------|------------|--------|-------|
> | 0.025 (0.5x) | 77.83 | 43.35 | 0.1076 | 0.1849 |
> | **0.05 (1x, default)** | **77.47** | **43.43** | **0.1113** | **0.1785** |
> | 0.1 (2x) | 77.84 | 43.54 | 0.1050 | 0.1791 |
> | 0.4 (8x) | 77.97 | 41.01 | 0.1106 | 0.1845 |
> | 0.8 (16x) | 77.86 | 44.40 | 0.1035 | 0.1833 |
>
> **γ sweep** (fix β=0.05):
>
> | γ | J&F ↑ | AU-PAvPU ↑ | AURC ↓ | ECE ↓ |
> |---|-------|------------|--------|-------|
> | 0.1 (0.5x) | 77.72 | 42.77 | 0.1106 | 0.1741 |
> | **0.2 (1x, default)** | **77.84** | **42.62** | **0.1072** | **0.1775** |
> | 0.5 (2.5x) | 78.41 | 43.34 | 0.1047 | 0.1760 |
> | 1.6 (8x) | 78.03 | 43.29 | 0.1079 | 0.1878 |
> | 3.2 (16x) | 78.22 | 41.28 | 0.1093 | 0.1830 |
>
> Across 0.5x–16x for both, all metrics remain stable (J&F range ≤0.77, AURC range ≤0.008), confirming robustness to hyperparameter choices.
>
> ---
>
> ## Q5: Loss component ablation
>
> | Configuration | J&F ↑ | AU-PAvPU ↑ | AURC ↓ | ECE ↓ |
> |--------------|-------|------------|--------|-------|
> | **RUAC (full)** | **81.62** | **63.72** | **0.092** | **0.1172** |
> | w/o AUE (γ=0) | 79.87 | 58.90 | 0.102 | 0.1232 |
> | w/o KL | 81.26 | 62.43 | 0.0907 | 0.1141 |
>
> Removing AUE (= Bayes-SAM2 in the main table) causes the largest drop (J&F −1.75, AU-PAvPU −4.82), confirming adversarial calibration as the key contributor. KL has minor impact (J&F −0.36); it serves as a training stabilizer.
>
> ---
>
> ## Q6: MC sample analysis
>
> Same checkpoint, only inference-time N varies:
>
> | N | J&F ↑ | AU-PAvPU ↑ | AURC ↓ | ECE ↓ |
> |---|-------|------------|--------|-------|
> | 1 | 81.61 | 63.36 | 0.096 | 0.1147 |
> | 5 | 81.60 | 63.32 | 0.089 | 0.1159 |
> | **20 (default)** | **81.62** | **63.72** | **0.092** | **0.1172** |
> | 50 | 81.61 | 64.14 | 0.091 | 0.1191 |
> | 100 | 81.62 | 63.87 | 0.090 | 0.1171|
>
> J&F is constant since the forward pass uses analytic Weibull expectation; MC only affects the uncertainty map. AURC saturates at N≈5, AU-PAvPU is stable at N=1. Unlike MC Dropout or ensembles, our samples come from a single parametric posterior whose (λ, κ) already encode uncertainty.
>
> We will incorporate all above analyses into the revised manuscript.

---

> > ### Author Rebuttal · Reviewer_7xn4 · 2026-04-02
> >
> > Thank you for the detailed rebuttal and additional experiments. The response addresses part of my concerns, and I will update my score accordingly. The claim in Appendix A that predictive uncertainty cannot be additively decomposed into aleatoric and epistemic components appears to be factually incorrect or overstated, regardless of whether the paper adopts such a decomposition. There is also some inconsistency in the narrative, as the paper downplays AU/EU decomposition while building on BNDL, which is explicitly motivated by it.

---

> > > ### Author Response · Authors · 2026-04-02
> > >
> > > Thank you very much for your constructive comments and kind reminder regarding the AU/EU decomposition problem. We consider your reminder very important, because it reveals an issue we had previously completely overlooked, although this issue is only related to the background narrative.
> > >
> > > Based on your comments, we have thoroughly reviewed previous literature. You are absolutely right; although our method does not pursue AU/EU decomposition, our baseline BNDL is inspired by AU/EU decomposition. Our previous decomposition statement in Appendix A was indeed exaggerated. Therefore, we have decided to revise the corresponding section in Appendix A, removing the statement about the inability to decompose uncertainty. We will also provide **a comprehensive description of uncertainty estimation, AU/EU, BNDL, and their relationship to our method** at the beginning of Appendix A.
> > >
> > > Initially, we plan to revise it as follows:
> > >
> > > > Uncertainty quantification (UQ) aims to quantify the confidence in the correctness of predictions. From a Bayesian perspective (Gal & Ghahramani, 2015), uncertainty can be decomposed into AU (the inherent randomness in the data) and EU (the uncertainty inherent in the model itself). BNDL employs a two-stream architecture to capture AU and EU separately. For AU modeling, BNDL uses a Weibull distribution to model the latent representation z. For EU modeling, BNDL uses the weights of the final decision layer as stochastic latent variables.
> > > >
> > > > Our uncertainty estimation method is based on BNDL because it is a lightweight, SOTA UQ method and improves the interpretability of the neural network. This is crucial for our task (using an interpretable model to segment images under various zero-shot domain shifts, i.e., interpretably segment anything). However, we did not pursue a further accurate decomposition of AU/EU, because we believe that a more important issue for our task is _how to guarantee robust uncertainty-accuracy correlation under domain shift_ (an orthogonal issue to AU/EU decomposition). Existing uncertainty estimation methods, including BNDL, still assume a high correlation between the training and test data domains. However, this assumption is no longer valid given the current trend in fundamental models: SAM2's test data domain is highly variable, covering multiple dramatic shifts. Therefore, the performance improvement of BNDL+SAM2 in our experiments was extremely limited.
> > > >
> > > > This led to our research: Segment Anything with Robust Uncertainty-Accuracy Correlation (RUAC). Our RUAC method employs a bio-inspired uncertainty adversarial design and is **the first method that leverages uncertainty-based adversarial training to simultaneously improve SAM's segmentation accuracy and provide calibrated pixel-wise uncertainty estimation**.
> > >
> > > Thank you again for your key comments! We believe they not only enhance the rigor of our narrative, but also greatly clarify the significance of our research.

---

### Official Review · Reviewer_NoDi · 2026-03-13

**Soundness:** 2
**Presentation:** 2
**Significance:** 3
**Originality:** 2
**Overall Recommendation:** 3
**Confidence:** 4

**Summary:**

This paper addresses the issue that SAM-based methods tend to be overconfident and provide only mask-level confidence. Furthermore, the authors point out that even when existing uncertainty estimation methods are jointly trained, they tend to work only on the source domain used during training, while producing irrelevant errors in different target domains.

To address this problem, the paper proposes Robust Uncertainty-Accuracy Correlation (RUAC), aiming to ensure that uncertainty consistently reflects the actual prediction errors. To achieve this, the authors artificially generate domain shift scenarios during training using adversarial perturbations. Specifically, Style perturbation generates appearance changes through AdaIN-based style transformation, while Deformation perturbation creates shape variations through geometric distortions. The Bayesian mask decoder, corresponding to the min player, is trained so that uncertainty aligns well with prediction errors. Based on this framework, the paper shows improved results in both segmentation performance and uncertainty estimation compared to existing SAM-based methods.

**Compliance With Llm Reviewing Policy:**

Affirmed.

**Key Questions For Authors:**

Refer to the weakness, especially the necessity of stylization (or adding perturbation).

**Limitations:**

No limitation or failure case is stated in this work.

**Strengths And Weaknesses:**

**Strengths**

- Since SAM-based approaches have strong segmentation capabilities and are widely used across many applications, addressing the limitation that they only provide mask-level confidence is important. Predicting pixel-level uncertainty is also a meaningful direction.

- To alleviate the scale mismatch problem that occurs when the domain changes, the paper proposes a perturbation-based approach and performs minimax training together with a Bayesian mask decoder.

- The method shows meaningful improvements compared to the existing SAM2 approach and provides experimental evidence of its superiority and robustness through evaluations in various environments.

**Weaknesses**

- Fig.1 appears on the first page and occupies a large amount of space, but it does not seem sufficiently informative to justify that space.

- Moreover, the proposed techniques are not particularly novel. In essence, the method injects noise only into specific regions based on the ground truth and trains the decoder so that higher uncertainty scores are produced in those regions. The Style/Deformation Networks can be considered as just one of many possible ways to realize this idea. In other words, it is not entirely convincing that the proposed approach is the best way to implement this idea. It would be necessary to verify whether the proposed method actually performs better compared to simple but strong baselines such as existing adversarial attacks or random noise injection.

- It would also be beneficial to provide additional evaluation metrics such as ECE.

---

> ### Author Rebuttal · Authors · 2026-03-29
>
> We sincerely appreciate your time and effort in reviewing. We are glad that you recognized the importance of pixel-level uncertainty for SAM. Below are our responses.
>
> ---
>
> ## W1: Fig. 1
>
> Thank you for raising this concern. We want to clarify the intended message of Fig.1.
>
> 1. We want readers to quickly see the accuracy and interpretability gains of our method, as is commonly done in image generation works (InstaFlow [Liu et al., ICLR 2024], YOSO [Luo et al., ICLR 2025], GigaGAN [Kang et al., CVPR 2023]).
> 2. Fig.1 serves as a teaser communicating three keys of our work:
>     - **Cross-domain evaluation setting.** Both the upper part Fig. 1a and lower part Fig. 1b share the same diverse out-of-domain test images spanning 4 categories, showing that our evaluation tests real-world generalization rather than in-domain performance.
>     - **The problem: SAM2's Mask-level Confidence Confusion (MCC).** Fig. 1a shows that vanilla SAM2, trained on a *fixed* domain, produces **unexplainable** segmentation and **confused** confidence maps on various OOD domains. This is the core motivation of our work.
>     - **RUAC's solution.** Fig. 1b shows that RUAC, by making the training domain *dynamic* via adversarial perturbations and a Bayesian Mask Decoder, produces **interpretable** segmentation and **robust** uncertainty maps across all four OOD domains.
>
> We sincerely appreciate your feedback on improving the presentation. We will optimize the layout in the revised version.
>
> ---
>
> ## W2: Novelty
>
> Thank you for your valuable feedback. Our novelty lies in four aspects:
>
> ### (1) RUAC Framework
>
> SAM's segmentation quality and interpretability still need improvement in real-world OOD scenarios, but retraining from scratch is expensive, and adaptation methods (e.g., LoRA [Hu et al., ICLR 2022], HQ-SAM [Ke et al., NeurIPS 2023]) require per-domain data and risk catastrophic forgetting (SUM [Liu et al., NeurIPS 2024]). We propose RUAC, which is **the first method that leverages uncertainty-based adversarial training to simultaneously improve SAM's segmentation accuracy and provide calibrated pixel-wise uncertainty estimation**. Specifically:
>
> - **vs Foundation model and uncertainty methods**: Recent SAM adaptation methods (HQ-SAM [Ke et al., NeurIPS 2023], EfficientSAM [Xiong et al., CVPR 2024], H-SAM [Cheng et al., CVPR 2024]) improve accuracy but **provide no interpretability**. While uncertainty methods (UncertainSAM [Kaiser et al., ICML 2025], Bayes-SAM2 (BNDL [Hu et al., ICLR 2025] + SAM2)) add interpretability, they **do not improve accuracy**. RUAC **simultaneously improves both accuracy and interpretability** across 23 OOD datasets (Table 1).
> - **vs FT methods (SUM [Liu et al., NeurIPS 2024])**: FT methods require per-domain fine-tuning with domain-specific data and cannot improve generalization across multiple domains simultaneously. RUAC requires no target-domain data and improves both segmentation and calibration across all 23 OOD domains.
>
> ### (2) Pixel-level Uncertainty
>
> We adapt BNDL [Hu et al., ICLR 2025] from image-level to **pixel-level** uncertainty within SAM2's mask decoder. BNDL originally produces a single uncertainty value per image. We extend it with spatially-varying uncertainty prediction and a dual-path architecture for SAM2's multi-mask output.
>
> ### (3) Adversarial Implementation
>
> Existing adversarial attacks (e.g., PGD) generate **unconstrained** pixel-space perturbations that deviate from realistic distributions. Our approach constrains perturbations to two biologically-motivated dimensions, **style** (appearance/texture) and **shape** (geometric), ensuring plausible distribution shifts:
>
> - For style, we propose **Style adversarial network**, which builds upon AdvStyle [Zhong et al., NeurIPS 2022] and extends it to **collaborative perturbation on multi-object styles**.
> - More importantly, we propose a **Deformation adversarial network**. Geometric deformation as adversarial perturbation remains **largely underexplored**. Our successful integration demonstrates its effectiveness, and we hope this encourages further investigation.
>
> Following the suggestion, we compare against two baselines sharing the same decoder and loss:
>
> | Method | J&F ↑ | AU-PAvPU ↑ | AURC ↓ | ECE ↓ |
> |--------|-------|------------|--------|-------|
> | Random Noise | 80.81 | 52.24 | 0.112 | 0.1259 |
> | PGD | 79.90 | 52.89 | 0.115 | 0.1295 |
> | **RUAC (Ours)** | **81.62** | **63.72** | **0.092** | **0.1172** |
>
> Random Noise and PGD achieve AU-PAvPU 52.24, RUAC reaches 63.72. Unconstrained perturbations fail to expose the calibration-relevant failure modes that our perturbations capture. We will add this comparison in the revision.
>
> ---
>
> ## W3: Evaluation
>
> We have included ECE in our supplementary material (Section J.1, Table S6). RUAC achieves 0.117 ECE↓ vs Bayes-SAM2's 0.123.
>
> We will include ECE in the main paper in the revised version.
>
> If you have further questions, please don't hesitate to let us know!

---

> > ### Author Rebuttal · Reviewer_NoDi · 2026-04-03
> >
> > Thank you for the response. Some of my concerns (e.g, ECE) are addressed. However, still I believe the novelty is limited, while the motivation is good. In short, the necessity of the proposed perturbation design remains unclear.
> > While the authors compare against random noise and PGD, these baselines are insufficient to justify the use of the proposed **structured** perturbations. PGD/Random noise is actually one of possible baseline, not the strong experiment to support the necessity of the structured perturbation.  Comparisons against stronger and more relevant baselines (e.g., color jitter, non-adversarial style transfer, or region-level perturbations) are required.

---

> > > ### Author Response · Authors · 2026-04-06
> > >
> > > We thank the reviewer for the constructive feedback. We agree that comparisons against stronger baselines are important to justify the necessity of our perturbation design. We have conducted the requested experiments and provide both empirical results and a mechanistic analysis below.
> > >
> > > ---
> > >
> > > ### $\color{red}{\textbf{Additional Baseline Comparisons}}$
> > >
> > > Regarding color jitter: it is already included as a standard data augmentation in **all** methods' training pipelines, so it serves as a shared baseline rather than a separate comparison.
> > >
> > > Following the reviewer's suggestion, we compared our RUAC to a strong **diffusion-based baseline StyleGen** we constructed following recent diffusion-based augmentation methods (ALIA [Dunlap et al., NeurIPS 2023], DA-Fusion [Trabucco et al., ICLR 2024]) using SD 1.5 img2img. Besides, we also compared RUAC to feature-statistics augmentation (MixStyle [Zhou et al., ICLR 2021], DSU [Li et al., ICLR 2022]), and region-level perturbation (Patch [Nesti et al., WACV 2022]). The results are following:
> > >
> > > | Method | J&F ↑ | AU-PAvPU ↑ | AURC ↓ | ECE ↓ |
> > > |--------|-------|------------|--------|-------|
> > > | Random Noise | 80.81 | 52.24 | 0.112 | 0.1259 |
> > > | PGD [Madry et al., ICLR 2018] | 79.90 | 52.89 | 0.115 | 0.1295 |
> > > | Patch [Brown et al., 2017; Nesti et al., WACV 2022] | 80.60 | 56.29 | 0.105 | 0.1311 |
> > > | MixStyle [Zhou et al., ICLR 2021] | 80.37 | 55.46 | 0.100 | 0.1334 |
> > > | DSU [Li et al., ICLR 2022] | 80.35 | 58.02 | 0.097 | 0.1251 |
> > > | StyleGen v1 (2 prompts) [Dunlap et al., NeurIPS 2023; Trabucco et al., ICLR 2024] | 79.33 | 55.10 | 0.101 | 0.1363 |
> > > | StyleGen v2 (9 prompts) [Dunlap et al., NeurIPS 2023; Trabucco et al., ICLR 2024] | 78.19 | 48.87 | 0.111 | 0.1720 |
> > > | **RUAC (Ours)** | **81.62** | **63.72** | **0.092** | **0.1172** |
> > >
> > > RUAC outperforms all baselines on every metric. All methods evaluated on 23 OOD datasets. We also would provide StyleGen prompt details in supplementary materials.
> > >
> > > ---
> > >
> > > ### $\color{red}{\textbf{Why RUAC's Structure Perturbation Is Necessary}}$
> > >
> > > We thank the reviewer for pushing us to go beyond empirical comparisons.
> > >
> > > Unlike classification, segmentation requires per-pixel labels (i.e. masks), which makes existing approaches limited: (1) non-adversarial methods cannot identify network-specific hard data; (2) diffusion-based synthesis risks semantic layout drift; (3) the region perturbation methods simply mixes patches and cannot naturally generate targeted structural changes; and (4) style transfer/noising can perturb images but not masks. Thus, the key advantage of RUAC’s structured perturbation is **jointly perturbing pixels and masks**. This is especially useful for segmentation, where the challenge lies not only in appearance shifts but also in large shape variation. By perturbing both, RUAC encourages shape-aware, boundary-sensitive representations and improves robustness to style and structural changes. As a result, the network becomes more robust to both style changes and structural deformations, which are both pervasive in real-world segmentation tasks.
> > >
> > > Furthermore, we provide a detailed mechanistic explanation by analyzing how each method's training-time perturbation aligns with real OOD domain shift at the **per-channel** level of the mask decoder features. For each method, we extract 32-d foreground pixel features (pooled within dilated GT masks at stride-4 resolution) and compute:
> > > - **Augmentation shift**: per-channel mean difference between augmented and clean features on ID data (MOSE)
> > > - **OOD shift**: per-channel mean difference between OOD and ID features, averaged over 23 zero-shot datasets
> > >
> > > Next, we compute the Channel Alignment = Pearson correlation across 32 channels between augmentation shift and OOD shift. A high correlation means the method perturbs the channels that actually shift under real domain change.
> > >
> > > | Method | Perturbation Type | Channel Alignment (r) | Perturbation Norm (Magnitude) |
> > > |--------|-------------------|-----------------------|-------------------|
> > > | Random Noise | Random pixel noise | −0.009 | 0.016 |
> > > | StyleGen v2 | SD img2img style transfer | 0.575 | 0.092 |
> > > | DSU | Feature statistics (Gaussian) | 0.815 | 0.0037 |
> > > | MixStyle | Feature statistics (mixing) | 0.821 | 0.0044 |
> > > | **RUAC** | **Adversarial style+deform** | **0.811** | **0.046** |
> > >
> > > Effective robustness training requires both high **alignment** and sufficient **magnitude**, only RUAC achieves both (r=0.81, norm=0.046), without any target domain knowledge. See [Figure](https://anonymous.4open.science/r/ICML_rebuttal-DED6/channel_alignment.pdf).
> > >
> > > A deeper representation-level analysis of what is missing in SAM2's pretrained features is provided in our response to Reviewer JF4c.
> > >
> > > We thank the reviewer again for pushing us toward these stronger comparisons. We will incorporate the baseline results and mechanism analysis into the revised manuscript. Should the reviewer have any remaining concerns, we are happy to address them during the discussion period.

---

### Decision · Program_Chairs · 2026-04-30

**Decision:**

Accept (regular)

**Comment:**

The paper received mixed ratings after rebuttal and discussion, in which reviewer 7xn4 upgraded to the borderline accept rating in the final justification but did not officially update the rating in the form, resulting in 3 positive and 1 negative rating in the end. Initially, reviewers had concerns about technical novelty/motivation, evaluations (e.g., metric, sensitivity of parameters, ablation study), and some technical clarity. After the rebuttal, except for reviewer NoDi, other reviewers are satisfied with the feedback. The AC took a closer look at the paper, reviews, and rebuttal, and found that the main concerns from reviewers are addressed well (including the one for reviewer NoDi in the rebuttal thread). Therefore, the AC recommends the acceptance rating, while highly encouraging the authors to improve the current version accordingly and release the code for reproducibility.